# Bigger, Regularized, Categorical: High-Capacity Value Functions are Efficient Multi-Task Learners

Michal Nauman[1,2]     Marek Cygan[2,3]     Carmelo Sferrazza[1]     Aviral Kumar[4]

Pieter Abbeel[1,5]

## Abstract

Recent advances in language modeling and vision stem from training large models on diverse, multi-task data. This paradigm has had limited impact in value-based reinforcement learning (RL), where improvements are often driven by small models trained in a single-task context. This is because in multi-task RL sparse rewards and gradient conflicts make optimization of temporal difference brittle. Practical workflows for generalist policies therefore avoid online training, instead cloning expert trajectories or distilling collections of single-task policies into one agent. In this work, we show that the use of high-capacity value models trained via cross-entropy and conditioned on learnable task embeddings addresses the problem of task interference in online RL, allowing for robust and scalable multi-task training. We test our approach on 7 multi-task benchmarks with over 280 unique tasks, spanning high degree-of-freedom humanoid control and discrete vision-based RL. We find that, despite its simplicity, the proposed approach leads to state-of-the-art single and multi-task performance, as well as sample-efficient transfer to new tasks.

## 1  Introduction

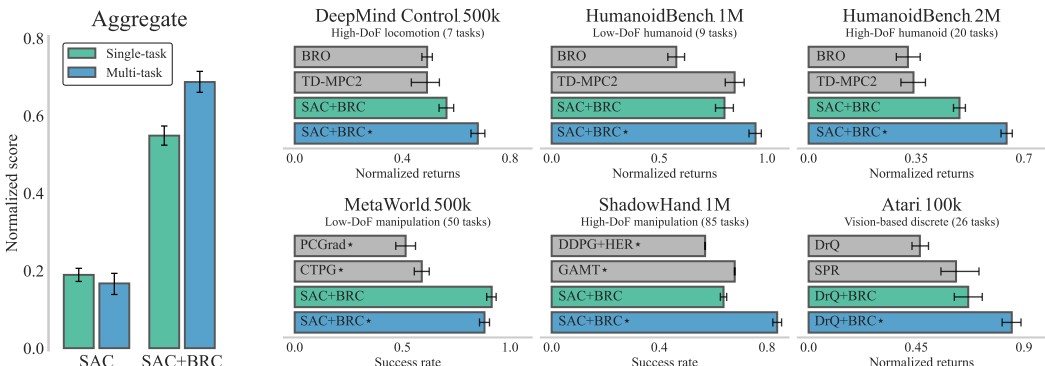

Figure 1: **Scaling multi-task training leads to state-of-the-art performance.** Naïve scaling of SAC to multi-task decreases the aggregate performance (**left**). Our proposed method (BRC) works both in single and multi-task learning and provides a pronounced performance improvement over previous approaches, including optimized single-task learners (**right**). We denote multi-task with ⋆ and share additional details in Appendix D.

Large-scale neural networks trained on large, diverse datasets have led to substantial advances in natural language processing [28, 18, 105] and computer vision [30, 78, 56]. These models, typically trained in a multi-task setting [85, 24, 1], exhibit not only strong performance across tasks seen during training but also demonstrate impressive sample efficiency when transferred to new domains or new

39th Conference on Neural Information Processing Systems (NeurIPS 2025).

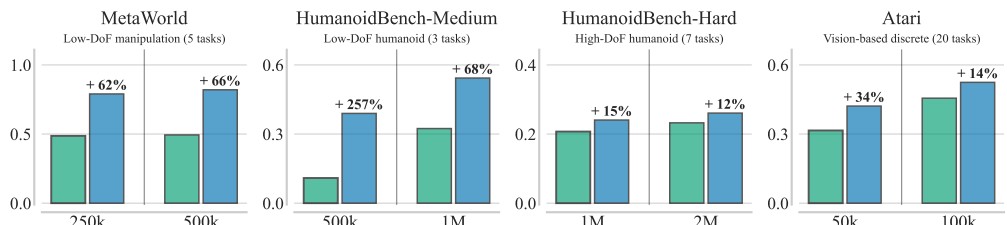

Figure 2: **Scaling multi-task training allows for sample-efficient transfer to new tasks.** We compare the performance of single-task BRC agent trained from scratch (green), to an agent initialized with our pretrained multi-task BRC agent trained on different tasks (blue). We find that transferring a multi-task BRC model to new tasks leads to better sample efficiency than learning from scratch. Y-axis denotes the average final performance.

tasks, including those that these models were never trained on [1, 102, 105, 66]. This combination of generalization and transfer has become the cornerstone of modern machine learning, suggesting that scaling models and data in a unified training regime can yield powerful and versatile systems.

Despite growing interest in reinforcement learning (RL) [25, 27, 79, 53], little is known about how RL scales [89]. This is particularly true for online RL algorithms, where recent advances have mainly focused on scaling model capacity for learning with limited single-task data [91, 77], making it difficult to assess how these methods perform when the variety of tasks and data increases [104, 46]. Addressing this gap is especially important, as experience with large language models shows that genuine scaling gains arise only when model capacity is paired with sufficiently diverse data [52, 45, 47, 69]. Unfortunately, previous work shows that naïvely extending value-based methods to multi-task RL is far from straightforward due to issues like reward imbalance [46] and gradient interference [120]. As such, practical systems often fall back on distillation from single-task experts [88, 65, 38, 21, 114, 111], offline learning from curated demonstrations [20, 7, 87, 17, 123, 119] or specialized techniques to resolve gradient conflicts [22, 92, 120, 23, 67]. As such, online training of high-capacity value-based agents on many tasks remains a significant open challenge.

We demonstrate, for the first time, that value models trained with online temporal-difference learning scale to the billion-parameter range, challenging the belief that such capacity requires offline datasets and behavioral cloning [98, 70]. We show that, similarly to supervised learning, combining model capacity with multi-task RL training leads to significant improvements over single-task oracles, as well as pronounced sample-efficiency gains when transferring to new tasks. The key design choices that unlock this behavior include significantly increasing critic capacity using normalized residual architectures, stabilizing temporal difference gradients through cross-entropy loss, and modeling task diversity via task embeddings instead of separate network heads. We run experiments using over 280 complex tasks from 5 benchmarks and find that the proposed combination of design choices produces simple agents that offer state-of-the-art performance in both single-task and multi-task scenarios, all while using a *single hyperparameter configuration*. Our contributions are as follows:

- **Model scaling in online RL** - we show design choices that allow for scaling value-based RL up to 1B parameters (Figures 4 & 12) without the use of offline data [61] or behavioral cloning [98]. We show that using such high-capacity models addresses the issues of gradient magnitudes [46] and conflicts [120] that are associated with multi-task RL (Figures 3 & 5).

- **Data scaling in multi-task RL** - we show that multi-task RL can yield improved sample efficiency and final performance as compared to single-task learning with state-of-the-art agents (Figure 1), despite using significantly less gradient updates (Figure 9). We find that, given common embodiment, agent performance scales with the number of tasks used in training (Figure 10).

- **Sample-efficient model transfer** - we find that initializing training from a pretrained, multi-task value model can lead to significant improvements in sample efficiency when compared to learning from scratch (Figure 2). We find the sample efficiency and final performance of the pretrained model transferred to new tasks is scaling with the capacity of the multi-task model, as well as the number of tasks used in pretraining (Figure 10).

Our goal is to show that, much like in vision and language modeling, larger models trained on more diverse streams of experience lead to policies that not only perform well in tasks used in multi-task learning but also transfer to new problems in a sample efficient manner. As such, we take a step toward bridging the methodology gap between supervised learning and value-based control.

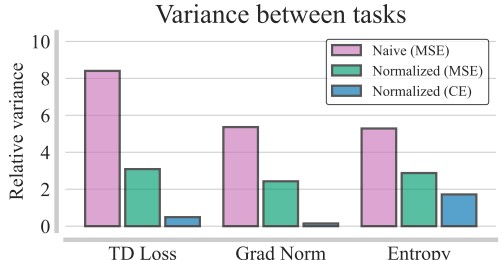 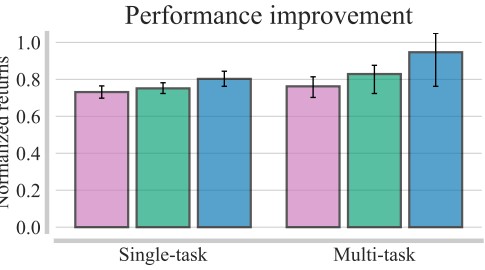

Figure 3: **Cross-entropy loss stabilizes online multi-task learning.** We investigate BRC with naive application of MSE loss (purple), MSE loss paired with return normalization (green) and cross-entropy paired with return normalization (blue) on HB-MEDIUM. Varying reward magnitudes in multi-task learning can destabilize learning of certain tasks, which translates to high variance of signals between tasks (**left**). Stabilizing this effect via cross-entropy loss allows for improved scaling when moving from single to multi-task learning (**right**).

## 2 Related Work

**High-capacity value models.** Previous work indicates that naïve scaling of model capacity in single-task RL leads to divergent behavior when training via temporal difference loss [15, 91, 77]. Interestingly, it was shown that using layer normalization [77, 63], feature normalization [59, 61], weight normalization [80], batch normalization [13, 19] or classification losses [40, 41, 33] can stabilize that divergent behavior and improve performance when using high-capacity models. Whereas these works investigated the above design choices in isolation, we explicitly combine normalized residual Q-value architectures [77] with cross-entropy loss via categorical Q-learning [11] and show that using these is crucial for efficient scaling in multi-task RL.

**Multi-task learning.** Previous work showcased a variety of issues with multi-task RL, such as divergence due to heterogeneous reward scales [104, 46] or conflicting gradients [120, 67] and proposed solutions such as reward [46] or gradient [22, 116] normalization, gradient projection [120] or distillation of single-task expert policies [104, 114, 111]. In contrast to these works, we show that increasing model capacity paired with training via cross-entropy loss is very effective at stabilizing multi-task learning, leading to policies that significantly outperform state-of-the-art single-task oracles. Furthermore, previous work considered multi-task training with high-capacity models [61, 41, 98]. In these works, the authors focus on offline learning from fixed datasets containing high-return trajectories. In contrast to these works, we focus on online multi-task learning without expert data.

**Transferable generalist policies.** Prior work investigated the possibility of learning a generalist agent that can solve many tasks and transfer to new ones [87, 17, 41, 103]. Due to multi-task RL complexities, many works train generalist agents via behavioral cloning on expert demonstrations [87, 31, 117]. These works often use pre-trained vision and language backbones [103, 68, 48] and achieve transfer via fine-tuning the pretrained model weights [54] or conditioning the pretrained model on text or visual task encoding [17, 123]. Although we are also interested in learning transferable generalist policies, our work assumes non-stationary learning of value functions via temporal difference [100]. More RL-based approaches facilitate transfer by learning a task-agnostic encoder and retraining a linear layer when the task changes [9, 115, 32, 2] or appending a learned task embedding to the model inputs, with heuristic selection of embeddings for new tasks [34, 43, 86, 97]. Whereas our work also considers task embeddings, we learn these online by backpropagating the temporal difference loss.

## 3 BRC Approach

In this section, we outline the Bigger, Regularized, Categorical (**BRC**) approach and its design principles that, when combined, allow effective scaling of online value-based agents (Figure 6). We discuss these design choices below and share additional details in Appendix C.

**Cross-entropy loss via distributional RL.** Recent studies have shown that reframing regression problems as classification tasks can significantly improve performance in both supervised learning [106] and RL [33]. The Mean Squared Error (MSE) loss is known to depend on the scale of predicted values and is susceptible to outliers [14]. Note that this susceptibility is particularly problematic in multi-task learning, as differences in reward magnitudes introduce an implicit task prioritization:

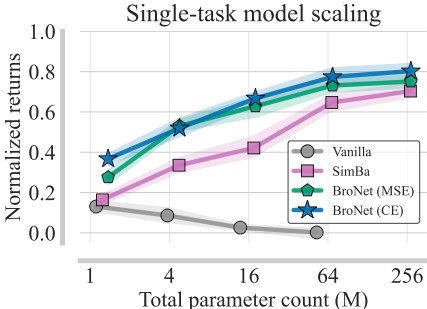
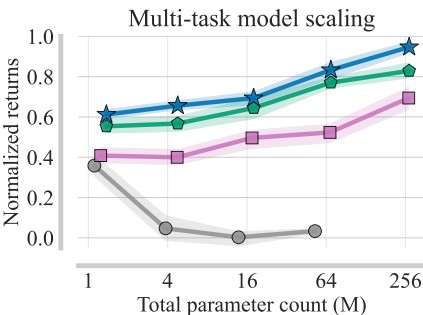

Figure 4: **BroNet paired with cross-entropy loss scales in both single and multi-task RL.** We compare scaling behavior of different architectures in single (**left**) and multi-task (**right**) when solving the HB-MEDIUM benchmark. We pair SAC with the vanilla [39], SimBa [63], BroNet with mean squared error loss [77], and proposed BroNet with cross-entropy loss architectures. Both figures report final performance after 1M steps.

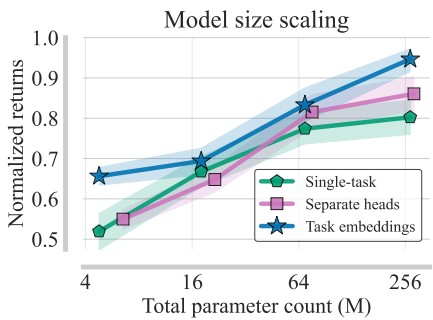
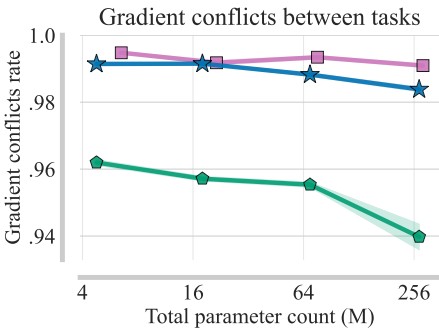

Figure 5: **Using task embeddings is preferable to separate heads design.** We compare the performance (**left**) and gradient similarity [120] (**right**) of different approaches for multi-task learning on HB-MEDIUM. We consider single-task, multi-task via separate heads [46, 61] and via task embeddings variants of our proposed BRC. We find that the task embeddings design outperforms other variants at all considered model scales and, interestingly, the separate heads design performs better than single task oracle only past certain model scale.

tasks with higher rewards produce higher TD errors and stronger gradient signals [104, 22, 46, 116]. This in turn results in an increase in the variance of TD errors, gradient norms, and policy entropy between tasks, destabilizing learning. We demonstrate this phenomenon in Figure 3, which demonstrates substantial variability in loss magnitudes, gradient norms, and policy entropy across tasks for a multi-task learner. Although reward normalization [46] partially mitigates this issue, we find that adopting a cross-entropy loss via categorical Q-learning [11] effectively balances gradients and ensures consistent learning signals across all tasks. We normalize the rewards in each task by dividing them by the maximal Monte Carlo return achieved in a given task at that point in learning (see Appendix C for details). Finally, we note that previous work builds theoretical arguments on why cross-entropy loss might stabilize TD learning [112, 113].

**Scaled Q-value model.** Prior work proposed scaling the Q-value model in single-task RL by using regularized ResNets [15, 77]. In particular, the BroNet architecture [77] was shown to achieve state-of-the-art performance on a variety of control tasks. To this end, we combine this architecture with the cross-entropy loss, and we refer to this variant as BroNet (CE). As shown in Figure 4, we observe that the trends observed in single-task learning persist in multi-task scenarios. Similarly to single-task setup, vanilla architectures experience performance degradation when the Q-value model size is increased, despite being exposed to more diverse data than in single-task learning. Furthermore, regularized architectures achieve steady performance improvements when increasing the capacity, with our proposed BroNet (CE) yielding the best performance in both single and multi-task setups. Furthermore, previous work found that online multi-task learning leads to gradient conflicts that substantially decrease the learner performance [120, 67]. Interestingly, we find that the rate of gradient conflicts decreases as the number of parameters increases (Figure 5), showing that complex interventions such as gradient surgery might not be required after a certain model scale. We discuss the considered architectures used to represent Q-values in Appendix C, as well as the definition of gradient conflicts in Appendix D.

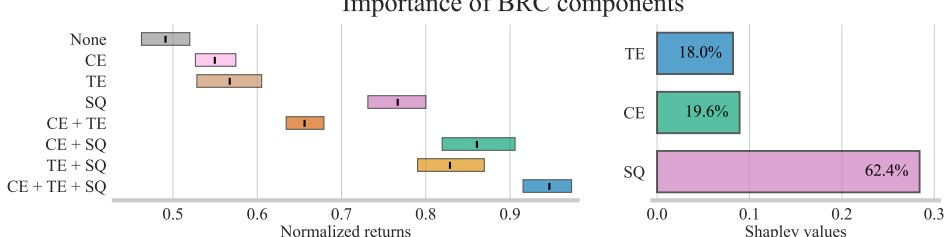

Figure 6: **Scaled Q-value model is the most important design choice.** (**Left**) We investigate the final performance of the base model paired with various combinations of scaled Q-value model (SQ), cross-entropy loss via categorical RL (CE), and learnable task-embedding module (TE) on HB-MEDIUM. (**Right**) SQ is associated with the highest Shapley value, accounting for 62% of improvements.

**Task embeddings.** To accommodate scalable learning across numerous tasks, we employ learnable task embeddings [86, 41]. By providing task identifiers and using task embeddings effectively transforms the multi-task setting into single-task by combining multiple MDPs into one [34]. In this unified MDP, the state representation is augmented with the learned embeddings, thereby allowing the models to distinguish which task it is solving. In contrast to approaches such as separate value and policy heads [120, 121, 61], learnable embeddings facilitate the discovery of shared structure between tasks, positioning similar tasks closer within the embedding space, without assumption that the Q-values of different tasks are linear with respect to the shared representation [81, 46, 2]. Crucially, these approaches do not explicitly supervise the learning of shared structure between tasks. In con-

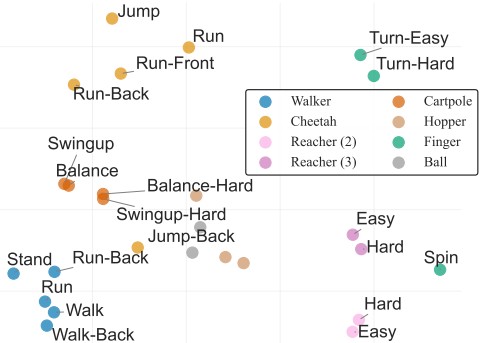

Figure 7: **Learnable task embeddings discover the underlying dynamics structure.** We graph the first two principal components of the task embeddings learned online on MW+DMC and find that the discovered embeddings cluster similar embodiments.

trast, we optimize embeddings by backpropagating the cross-entropy temporal difference (TD) loss, thus encouraging embeddings that represent well the value structure. As shown in Figure 7, our approach learns embeddings that cluster similar tasks. Furthermore, as shown in Figure 5, using task embeddings performs better than separate heads at various model scales. We detail our approach to using task embeddings in Appendix C.

**Summary.** The design choices that enable us to scale multi-task RL agents are: (**1**) using a scaled and regularized residual network architecture to represent the Q-values, (**2**) using cross-entropy instead of MSE loss to train the critic, and (**3**) conditioning models on task embeddings which are learned by backpropagating the temporal difference loss. We refer to the resulting approach as Bigger, Regularized, Categorical (**BRC**). We apply BRC to two popular model-free algorithms for continuous and discrete control, and study their scaling and transfer behavior. For continuous control experiments, we utilize Soft Actor-Critic (SAC) [39], whereas for discrete vision-based control, we use DrQ-$\epsilon$ [58] (we discuss hyperparameters in Appendix F). As shown in Figures 3 and 5, BRC tackles the problems of stability [104, 22, 46, 116] and gradient conflicts [120, 67] in multi-task RL. To assess the importance of BRC components, we evaluate the performance of every combination of these choices. As shown in Figure 6, we find that combining the above techniques leads to synergy effects, where removing any block always leads to a decrease in performance. Finally, we calculate the exact Shapley values [94] associated with each component and find that using the scaled Q-value model is the most important factor, accounting for 60% of the presented improvements.

## 4 Experimental Evaluation

We now discuss experiments designed to evaluate the effectiveness of our BRC approach in scaling with data and compute. Our goal is to answer the following questions: (**1**) How does our proposed method enhance the effectiveness of model capacity scaling in multi-task RL? (**2**) Does the perfor-

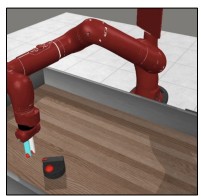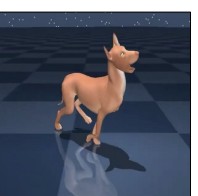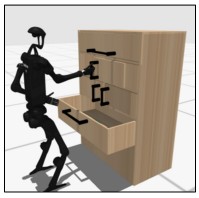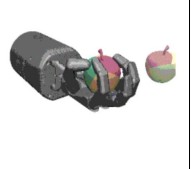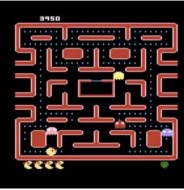

Figure 8: **We consider 283 tasks from 5 simulation benchmarks.** We test our approach with SAC [39] on MetaWorld, DeepMind Control, HumanoidBench and ShadowHand, and with DrQ-$\epsilon$ [58] on vision-based Atari. In both approaches we use single set of hyperparameters across all tasks and multi-task configurations.

mance of our multi-task learner improve when scaling the number of tasks? **(3)** Do the representations learned in online multi-task learning transfer to new tasks and result in more efficient learning than fresh initialization? **(4)** How does model capacity scaling impact the task-scaling and transfer capabilities of agents? We answer these questions through experiments described in the subsections below and analyzed in Section 5. For more details on our experiments see Appendix D.

**Benchmarks.** We consider a wide range of tasks, with a total of 283 diverse, complex control problems spanning five domains: DeepMind Control (DMC) [101], MetaWorld (MW) [121], HumanoidBench (HB) [93], Atari [10], and ShadowHand (SH) [49]. The tasks considered include locomotion and manipulation, different embodiments, numerous degrees of freedom ($|A|$ reaching 60 dimensions), fully sparse rewards, and vision-based control. We use the following multi-task configurations: DMC-HARD with dog (4 tasks) and humanoid (3 tasks) locomotion tasks trained for 500k environment steps per task [77], MW with 50 manipulation tasks for a Franka robot [121] trained for 500k environment steps per task, HB-MEDIUM with 9 humanoid locomotion tasks trained for 1M environment steps per task, HB-HARD with 20 whole-body locomotion and manipulation tasks for a humanoid equipped with Shadow Hands [93] trained for 2M environment steps per task, and SH with high DoF Shadow Hand manipulating 85 diverse shapes with fully sparse rewards [49] trained for 1M environment steps per task. We also consider an 80-task MW+DMC with multiple robot embodiments considered in Hansen et al. [41]. For vision-based experiments, we consider 26 tasks from Arcade Learning Environment (ATARI) [51]. Finally, we consider an additional 64 tasks from all benchmarks for transfer experiments. We list all the task sets considered in Appendix E.

**Training.** We compare our approach against a variety of modern single and multi-task approaches. We consider TD-MPC2 [41], DreamerV3 [40], BRO [77], SAC [39], TD7 [37], SimBa [63], SimBaV2 [64], MH-SAC [121], PCGrad [120], PaCo [99], CTPG [44], GAMT [49], DDPG+HER [5, 6], DrQ-$\epsilon$ [58], SPR [90], SimPLe [51], and DER [108]. Whenever possible, we show previously reported results, otherwise we run official repositories associated with the given method. For our approach, we use a single set of hyperparameters across all benchmarks, in both continuous and discrete action experiments. Specifically, we use hyperparameter values proposed in previous works for our backbone algorithms ([77] for SAC and [3] for DrQ-$\epsilon$). We use the 250M variant of the single and multi-task BRC model except when directly stated otherwise. In both cases, following prior works [41], we increase the batch size 4× to accommodate multi-task learning. In contrast to previous works [41, 64], we do not use learning rate scheduling or discount heuristics for our method. Furthermore, all multi-task agents interact with all tasks in parallel and keep updates-per-data (UTD) fixed at UTD = 2 [77]. As such, the multi-task learners perform fewer gradient steps than their single-task counterpart (e.g. with 20 tasks, the multi-task model performs 20 times fewer updates while performing the same number of environment steps). We detail hyperparameters in Appendix F.

**Transfer learning.** In our transfer experiments, we evaluate three adaptation protocols inspired by previous work. Firstly, we consider *model transfer* by initializing a single-task learner with the weights of the pretrained multi-task agent and continue learning only on data from the new task [61, 41]. Secondly, in what we call *embedding-tuning transfer*, we keep the pretrained parameters frozen [34, 60] and adapt only through a low-dimensional task embedding inferred from a less than a thousand target-task transitions. Finally, as a form of a baseline in transfer approaches, we consider an approach leveraging prior data [8, 122] in which we initialize the experience buffer of a fresh single-task learner with transitions stemming from multi-task pretraining (we refer to this approach as *data transfer*). In all cases, the multi-task model is not trained on the transfer tasks, mimicking the train-test split used in supervised learning [14]. We report results for transfer experiments in Figures 2, 10 and 11. We list the tasks used in multi-task and transfer learning in Appendix E.

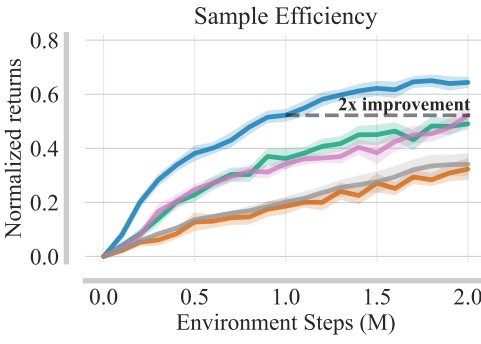
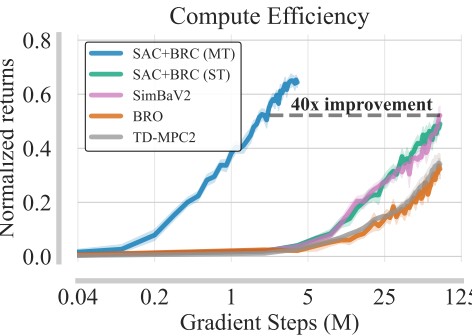

Figure 9: **Multi-task learning is sample and compute-efficient.** We compare the performance of our proposed single and multi-task BRC agents on HB-HARD benchmark and find that multi-task learning leads 2× improvement in sample efficiency (**left**) while performing 40× less gradient updates (**right**) when compared to SOTA single-task learners. This result shows that multi-task training can be very efficient.

**Evaluation.** Since tested environments use different reward scales, we report performance normalized to $[0, 1]$ according to practices drawn from previous works: for DMC, we divide the returns by 1000 [77]; for MW, we report success rates [121]; for HB-MEDIUM and HB-HARD we divide the returns by their success score [93, 63]; and for ATARI we divide the returns by scores achieved by humans [72]. When reporting aggregate, we report the sample mean across tasks with 95% confidence interval calculated via bootstrapping over seeds [3]. We detail the implementations in Appendix F, normalization values in Appendix E and report the unnormalized scores in Appendix H.

## 5 Analysis

This section summarizes the results of our experiments with the BRC approach, particularly highlighting its scaling and transfer properties. We adhered to the setup discussed in Section 4.

**Efficiency of multi-task learning.** Perhaps most importantly, we demonstrate that given the design choices outlined in Section 3, a single value model trained jointly multi-task setup not only matches but *surpasses* strong single-task specialists, establishing a clear generalist advantage in value-based RL (Figure 1). This result fills the gap left by recent work reporting parity with single-task oracles [87, 111]. Furthermore, as shown in Figure 9, the performance improvements are complemented by significantly lower compute load, with our multi-task model performing 40× fewer gradient updates to achieve the performance of the best single-task model. Since the multi-task model learns from all tasks in parallel, it uses more data than any individual single-task learner – we find that the outcome is consistent with observations in supervised learning, where data and compute can be traded off to reach a given performance level [71, 52, 47]. We note that BRC achieves this level of performance while using a single set of hyperparameters across all tasks proposed by prior work and tuned for single-task DMC [77]. We present detailed results, including training curves, in Appendix H.

**Scaling model capacity.** Similarly to single-task learning, increasing the model capacity can lead to substantial performance improvements in the multi-task setting (Figure 4). In fact, we observe that the increased data diversity resulting from multi-task learning allows the performance to scale beyond that of single-task RL - whereas in single-task the performance saturates at around 64M parameters, the multi-task model still improves beyond 250M parameters (Figures 4 and 12). Furthermore, as shown in Figure 5, we observe that increasing the parameter count reduces the rate of gradient conflicts for both single and multi-task learners, suggesting that high-capacity models can at least partly substitute interventions such as gradient surgery. To this end, as shown in Figure 1, we observe that our method outperforms multi-task specific approaches like CTPG [44], PaCo [99] or PCGrad [120]. Finally, we find that scaling the Q-value model is the most important design decision, accounting for more than 60% of the improvements of our approach according to the Shapley values (Figure 6).

**Scaling number of tasks.** We find that the BRC performance scales with the number of tasks used in pretraining, with the multi-task variant performing equally or better than single-task counterpart (Figure 1). Interestingly, we also observe that our proposed method trains robustly even when the number of robot embodiments grows (Figure 12). Furthermore, we find that given the common

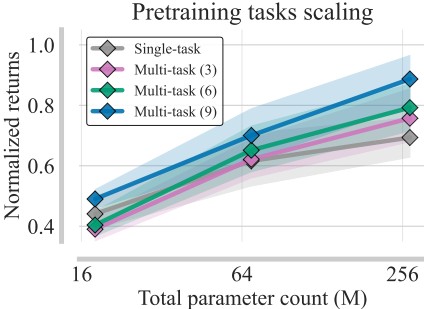
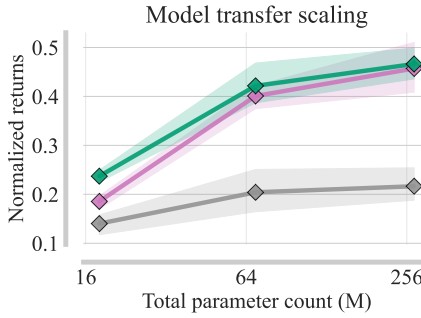

Figure 10: **Increased data and compute improves pretraining and transfer performance.** We train our multi-task agent on HB-MEDIUM using 3, 6 or 9 tasks for 1M environment steps. We find that the final performance on 3 shared tasks improves when increasing the number of pretraining tasks, with the trend most pronounced for the biggest evaluated models (**left**). We also investigate the final performance of the same models when transferring to 3 new tasks. The transfer performance is improving with both model capacity and the number of pretraining tasks (**right**). We detail the setup in Appendix D.

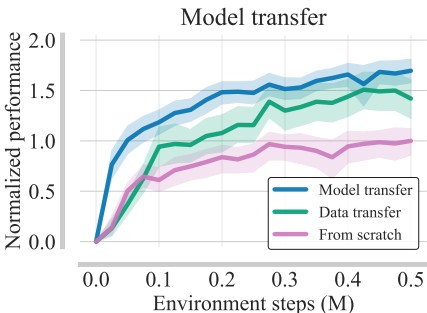
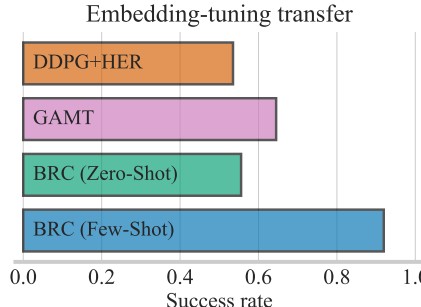

Figure 11: **Value-based RL agents are transferable.** (**Left**) We compare the performance of model transfer with the BRC agent to a fresh agent initialized with the buffer used in pretraining of the transferred BRC model (data transfer). We consider 15 transfer tasks from MW, HB-MEDIUM and HB-HARD listed in Appendix E.2. (**Right**) We also inspect the potential of fine-tuning solely via adjusting the task embedding, while keeping the policy and value models frozen (*embedding-tuning transfer*). We report success rates on 29 transfer tasks from the SH manipulation benchmark listed in Appendix E.2. We consider using random embeddings from pretraining tasks (zero-shot) and choosing the best-performing embedding from the pretraining set (few-shot). Surprisingly, the few-shot embedding transfer achieves over 90% success when manipulating new objects.

embodiment, pretraining using more tasks leads to both improved performance on pretraining tasks and improved capabilities when transferring to out-of-distribution tasks (Figure 10). This indicates that akin to supervised learning, increased data diversity in online RL leads to more general features that allow for sample-efficient transfer. Whereas our work showed empirically that such a scaling effect exists in high-DoF humanoid tasks, we believe that establishing a more comprehensive theory of task similarity in RL can be an impactful avenue for future research.

**Model transfer from multi-task RL** We observe that multi-task agents can learn features that transfer to new tasks, leading to improved sample efficiency and performance on various benchmarks (Figure 2). This transfer capability seems to be related to the size of the multi-task learner, with bigger models being better transfer learners (Figure 10). Examining the learning curves in Figure 11, we observe that *model transfer* not only outperforms learning from scratch but also converges more rapidly than the *data transfer* baseline. Although both methods rely on the same set of pretraining transitions, initializing with multi-task weights yields higher returns throughout training. Interestingly, as shown in Figure 11, the model transfer procedure (i.e. transferring weights of the pretrained model) is often more practical than data transfer (i.e. initializing a fresh agent with a replay buffer from multi-task pretraining [8]). This result is consistent with recent observations made in single-task offline-to-online fine-tuning [122]. We also note that the *data transfer* procedure differs from traditional offline-to-online RL [8, 122] or network resets [29] in that the *data transfer* method uses data generated when solving auxiliary tasks. Although the exact reason for the effectiveness of *model*

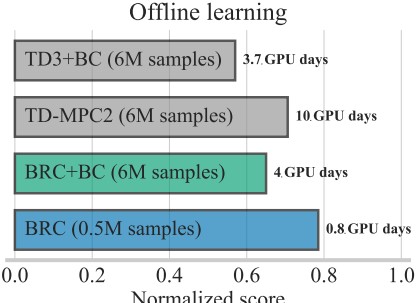
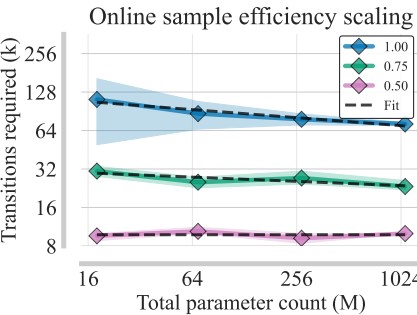

Figure 12: **Scaling online multi-task TD learning past 1B parameters.** We investigate the performance of online (BRC) and offline (BRC+BC) agents on the multi-task MW+DMC benchmark consisting of 12 different robot embodiments [41]. (**Left**) Online multi-task learning is significantly more compute and data-efficient than offline learning. (**Right**) We investigate the number of samples required for BRC agents to reach a percentage of offline TD-MPC2 final performance. The find that 1B BRC model is the most sample-efficient, reaching the performance of offline TD-MPC2 with 64k transitions, almost 100× less than TD-MPC2.

*transfer* requires further study, pretrained weights encode the dataset without inducing distribution mismatch when learning new tasks.

**Embedding-tuning transfer** We also investigate the possibility of transferring to new tasks by solely adjusting the task embeddings, while keeping the entire network frozen (embedding-tuning transfer). Here, we consider the Shadow Hand manipulation setup proposed in previous work [49], with pretraining focused on manipulating 85 distinct shapes, with evaluation on 29 out-of-distribution objects. As shown in Figure 11, using only 300 transitions for task embedding selection allows the BRC agent to achieve over 90% success rates when manipulating new shapes, outperforming specialized baselines using privileged information [49] or hindsight relabeling [5]. Furthermore, using a random embedding from the pretrained set yields satisfactory performance, showing that our multi-task policy conditioned on task embedding did not overfit to pretraining shapes. We believe that this behavior stems from the ability of the model to discover semantics of the underlying task space (Figure 7). This result shows that multi-task training with task-embeddings can lead to robust transfer and skill discovery just by manipulating the network input space, without updating the parameters of the policy or value model. We expand our analysis of embedding-tuning transfer in Appendix C.

**Multi-task value models learn more efficiently with online data** Recent work performed offline model-based training on 6M expert transitions per task from 80 MW and DMC tasks, containing 12 different embodiments [41]. We investigate the performance of BRC on this benchmark in both offline and online setup, where in the offline learning we use a behavioral cloning auxiliary objective following prior work [35, 82, 83] which we detail in Appendix C. As shown in Figure 12, offline multi-task RL remains challenging, even when using high-capacity value models, with online BRC performing nearly 20% better than its offline counterpart. Furthermore, in contrast to results in online learning (Figure 1), we find that the offline BRC agent slightly underperforms TD-MPC2, indicating that Q-learning with offline data might be harder than offline model-based learning. To better contrast the efficiency of offline and online multi-task value learning, we investigate the minimal number of online transitions required by BRC agents of different capacities (16M, 64M, 256M and 1B) to achieve a fraction (0.5, 0.75, and 1.0) of final TD-MPC2 performance on the MW+DMC benchmark. As shown in the right part of Figure 12, BRC sample efficiency still improves past 1B parameters, with logarithmic curves modeling the efficiency improvements well at different performance fractions.

## 6  Conclusions

Our study showed that online training of value functions by temporal difference loss scales beyond billion parameters and to many tasks. As discussed throughout the manuscript, scaled Q-value model trained via cross-entropy loss and conditioned on learnable task embeddings stabilizes the previously described issues of varying gradient magnitudes [104, 22, 46] and conflicting gradient [120, 67], resulting in state-of-the-art performance across a variety of tasks. Perhaps surprisingly, we found that scaled multi-task RL can be extremely computationally efficient, achieving the final performance of

single-task experts with 40× fewer gradient steps on the challenging whole-body humanoid control tasks. Finally, we showed that pretrained multi-task value models can be transferred to new tasks, improving both sample efficiency and final performance. We discuss the limitations of our approach in Appendix A. We hope that our work challenges the belief that "one agent per task" is the preferable workflow to training a scaled multi-task agent and provides a concrete foundation for online training of generalist value models in RL.

**Acknowledgments**

We also gratefully acknowledge the Polish high-performance computing infrastructure, PLGrid (HPC Center: ACK Cyfronet AGH), for providing computational resources and support under grant no. PLG/2024/017817. We also thank NVIDIA for providing compute resources through the NVIDIA Academic DGX Grant. Pieter Abbeel holds concurrent appointments as a Professor at UC Berkeley and as an Amazon Scholar. This paper describes work performed at UC Berkeley and is not associated with Amazon. Marek Cygan was partially supported by National Science Centre, Poland, under the grant 2024/54/E/ST6/00388. We would like to thank the Python [109], NumPy [42], Matplotlib [50], SciPy [110] and JAX [16] communities for developing tools that supported this work.

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

# A   Limitations

Our study is restricted to model-free setup and studying the effectiveness of online multi-task learning in model-based RL might be an interesting avenue for future research. Furthermore, in our discrete-action experiments we use the simple DrQ as base algorithm and our results in Atari100k are not competetive with state-of-the-art algorithms tuned for performance in Atari. As such, we believe that combining our insights with approaches like BBF or EfficientZero might be a promising avenue for further improvement of these state-of-the-art algorithms for Atari benchmark. We scaled both model capacity and task diversity and observed consistent performance gains, indicating that multi-task value-based RL benefits from larger networks and broader data. These results are encouraging but incomplete: they do not yet clarify when tasks help each other and when they do not. Our experiments suggest that tasks sharing the same embodiment usually reinforce one another, while mixed-embodiment suites show synergy in some cases and none in others. When synergy is absent, our method still learns, although at a slightly slower rate, whereas prior approaches often failed to converge. We believe that studying the conditions that create synergy between tasks in multi-task learning might be an important direction for future work. Finally, replay buffer memory usage is a limitation of our method and multi-task learning in general, with the memory scaling linearly with the number of tasks. In practice, we find the memory footprint to be manageable in most settings. Assuming Float32, on the HB-Medium benchmark with 9 tasks, the full replay buffer (1M transitions per task) occupies only 4.40 GB of memory.

# B   Background

We study an infinite-horizon Markov Decision Process (MDP) [12, 84], expressed by the tuple $(\mathcal{S}, \mathcal{A}, r, p, \gamma)$, where the state space $\mathcal{S}$ is continuous, while the action space $\mathcal{A}$ may be continuous or discrete. The function $r(s, a, s')$ gives the immediate reward received when action $a$ taken in state $s$ leads to the next state $s'$, and $p(s' \mid s, a)$ is the corresponding transition kernel. The scalar $\gamma \in (0, 1]$ is the discount factor. A policy $\pi(a \mid s)$ specifies a distribution over actions conditional on each state; its uncertainty is measured by the entropy $\mathcal{H}\big(\pi(\cdot \mid s)\big)$. Following Haarnoja et al. [39], we define the soft value in state $s$ as the expected discounted sum of rewards augmented by the policy entropies accumulated along the trajectory:

$$V^\pi(s) = \mathrm{E}_{a\sim\pi, s'\sim p}\left[r(s', s, a) + \alpha\mathcal{H}(\pi(s)) + \gamma V^\pi(s')\right],$$

The temperature parameter $\alpha \geq 0$ balances reward maximisation against exploration through entropy. The soft Q-value under a policy $\pi$ is:

$$Q^\pi(s, a) = \mathbb{E}_{s'\sim p(\cdot\mid s,a)}[r(s, a, s') + \gamma V^\pi(s')].$$

A policy is optimal when it maximizes the expected soft value of the initial state distribution, according to $\pi^\star = \arg\max_{\pi\in\Pi} \mathbb{E}_{s_0\sim p}[V^\pi(s_0)]$, with $\Pi$ the chosen policy class (for example, Gaussian policies). Soft values and soft Q-values are linked by $V^\pi(s) = \mathbb{E}_{a\sim\pi(\cdot\mid s)}[Q^\pi(s, a) - \alpha\log\pi(a \mid s)]$. In practice this expectation is often approximated with a single sample $a \sim \pi(\cdot \mid s)$. Off-policy actor–critic methods learn both the critic $Q_\theta$ and the actor $\pi_\phi$ from transitions $T = (s, a, r, s')$ stored in a replay buffer $\mathcal{D}$ [95, 73, 96]. The critic parameters minimize the squared soft temporal difference error $\theta \leftarrow \arg\min_\theta \mathbb{E}_{T\sim\mathcal{D}}\big(Q_\theta(s, a) - y(r, s')\big)^2$ with $y(r, s') = r + \gamma V_{\bar\theta}(s')$, where the target value

is computed with a slowly updated set of parameters $\bar{\theta}$ [107]. The policy parameters ascend the critic's estimate of the soft value $\phi \leftarrow \arg\max_\phi \mathbb{E}_{s\sim\mathcal{D},\, a\sim\pi_\phi(\cdot|s)}[Q_\theta(s,a) - \alpha \log \pi_\phi(a \mid s)]$. These updates iterate until the policy approaches the solution [55, 57].

## C    Bigger, Regularized, Categorical

In this section, we discuss the implementation details associated with the proposed BRC method.

### C.1    Base model

**Continuous action environments.** We use SAC [39] as the base model. Following previous work studying SAC sample efficiency [26, 74–76], we do not use clipped double Q-learning [36] while still using and ensemble of two critic networks, i.e. we update the critic networks according to the average output of the target networks, as opposed to taking the minimum. The proposed SAC+BRC approach is also related to the BRO algorithm. Conceptually, BRC can be understood as SAC without CDQL, with task embeddings and using BroNet critic.

**Discrete action environments.** We use DrQ-$\epsilon$ as our base model [118, 3]. Following Schwarzer et al. [91], instead of performing hard target updates, we update the target network every learning step using Polyak averaging.

### C.2    Cross-entropy loss via distributional RL

**Categorical RL.** We follow the categorical RL formulation as described in Bellemare et al. [11] As such, instead of predicting only the mean return, we can model the full distribution of returns. Following Bellemare et al. [11], we define the support as a discrete vector with $N_{\text{atoms}}$ values denoted by $z_i$, each defined by:

$$z_i = v_{\min} + \frac{(i-1)\big(v_{\max} - v_{\min}\big)}{N_{\text{atoms}} - 1}, \qquad i = 1, \ldots, N_{\text{atoms}}. \tag{1}$$

At time step $t$ the parametric approximation of the true return distribution is a function of $z_i$ and the probability mass assigned to each atom (denoted as $p_\theta(s_t, a_t)$). The critical observation is that return distributions obey a distributional Bellman equation [11]. The key implementation problem stems from the fact that categorical RL learns a distribution with fixed support. This is particularly problematic in a multi-task setting, where each environment can have vastly different reward scales, leading to different scales of Q-values. To remedy this issue, we rescale the rewards of each task according to a Monte Carlo approximation of the highest return encountered during the training.

**Return normalization.** The goal of reward normalization is twofold: firstly, we want to bound the returns such that there is no regret stemming from using categorical bounded Q-value representation; and secondly, we want to maintain a similar scale of returns between tasks. To this end, we maintain a per-task normalization factor that we denote $\bar{G}_i$ with $i = 1, \ldots, num\ tasks$. At a particular training step, the normalization factor is proportional to the maximal absolute returns observed until the given training step. To this end, whenever an episode concludes for any task, we calculate the Monte-Carlo returns for every state in that episode, and if the episode is truncated, bootstrap the Monte-Carlo estimate with the critic output. Then, we find the state with the biggest absolute value (i.e., $\max[|G_i(s_t)|]$ where $G_i(s_t)$ represents the Monte Carlo returns for state $s_t$). Finally, if the new return estimate is bigger than the current $\bar{G}_i$, we update it:

$$\bar{G}_i \leftarrow \max[\max[|G_i(s_t)|],\ \bar{G}_i] \tag{2}$$

We use $\bar{G}_i$ to normalize the returns during model updates: when we sample a new batch of data from the replay buffer, we normalize the returns according to:

$$r_i(s,a) = \frac{r_i(s,a) * V_{max}}{\bar{G}_i + \lambda_i} \tag{3}$$

Where $r_i(s, a)$ denotes the sampled rewards for the $i$th task, $V_{max}$ denotes the maximal returns bucket for categorical RL, and $\lambda_i$ corresponds to the maximum entropy correction for $i$th task, which we estimate via sum of a geometric sequence $\lambda_i = \alpha \mathcal{H}_i / (1 - \gamma)$, with $\gamma$ denoting the discount factor, $\alpha$ denoting the entropy temperature, and $\mathcal{H}_i$ denoting the empirical entropy of the policy conditioned on $i$th task embedding. Normalizing rewards according to Equation 4 guarantees that the modelled returns are bounded by $V_{max}$.

### C.3 Scaled Q-value model

Designing BRC, we built on previous work that studied scaling of Q-value models [61, 91, 77]. In this subsection, we detail the architectures used in our experiments.

**Proprioception-based control.** For proprioception-based control, we use the BroNet architecture. BroNet is an MLP with residual connections and layer normalization which we detail in Figure 13). As shown in Figure 4, we also tested SimBa [63], a follow-up work to BroNet that changes the initial layer normalization to a running statistics normalization, but found BroNet to perform noticeably better (Figure 4).

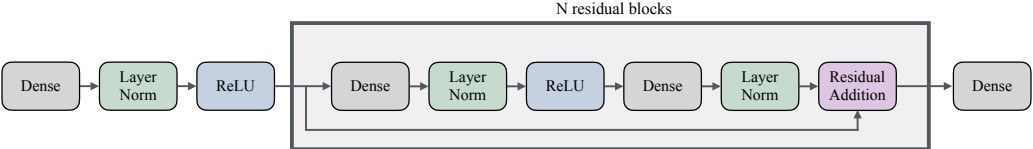

Figure 13: **BroNet used in the SAC+BRC method.** We use the exact residual architecture as presented in Nauman et al. [77].

**Image-based control.** In image-based experiments, we use the scaled Impala architecture [61, 91]. Following previous work [62, 77], we add a layer normalization between the encoder and Q-network modules. We show the architecture in Figure (14).

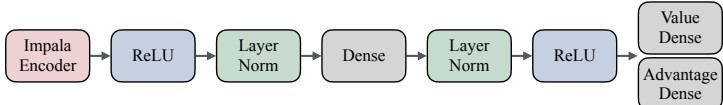

Figure 14: **Impala architecture used in the DRQ+BRC method.** The architecture is based on observations made in previous works [61, 91, 77].

### C.4 Task embeddings

We use an embeddings module[1] which we adjust by backpropagating the temporal difference loss. Following prior works [41], we constrain the L1 norm of the embeddings to be equal to 1 and leave experimentation with this setting for future work. We concatenate the states with task embeddings whenever we forward the actor or critic models according to Figure 15.

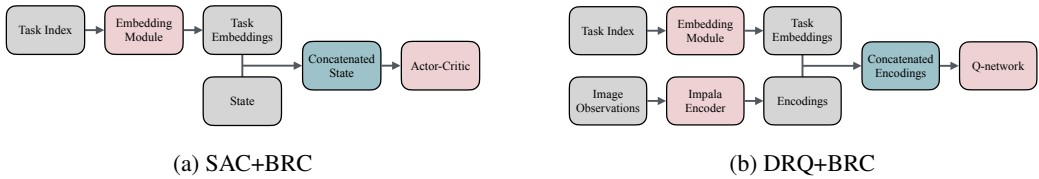

(a) SAC+BRC                                    (b) DRQ+BRC

Figure 15: **Our approach for concatenating observations with the learned task embeddings.** We use slightly different approach for proprioceptive (**left**) and vision-based (**right**) control. In particular, in vision-based tasks, we learn a task-agnostic encoder by concatenating task embeddings to the output of the encoder.

As shown in Figure 15, we use slightly different approaches for proprioceptive and vision-based control. In proprioceptive control, we simply concatenate the task embedding with the raw proprio-

---

[1]The embedding module is called `nn.Embed` in JAX and `nn.Embedding` in Torch.

ceptive state, so both the actor and critic are directly conditioned on the task. In Vision-based control, we first pass the RGB observation through the IMPALA encoder. We then concatenate the resulting image feature with the task embedding and feed the joint vector to the Q-network. The image encoder is therefore task-agnostic, while the Q-network is conditioned on both the visual features and the task.

# D   Details on Experiments

In this Section, we describe the procedure used to generate each figure, as well as discuss the baselines used in the manuscript. Unless explicitly stated otherwise, we use hyperparameter configuration described in Appendix F for both single and multi-task BRC variants.

Table 1: Description of model sizes shown in Figures 4, 5, 10 and 12.

| Model Size | Number of blocks | Block width |
|---|---|---|
| $\approx 1M$ | 2 | 256 |
| $\approx 4M$ | 2 | 512 |
| $\approx 16M$ | 2 | 1024 |
| $\approx 64M$ | 2 | 2048 |
| $\approx 256M$ | 2 | 4096 |
| $\approx 1B$ | 2 | 8192 |

## D.1   Figures

**Figure 1.** We report the average final performance on every considered benchmark. We take average with respect to random seeds (we use a minimum of 5 random seeds per method) and different tasks. We denote the training lengths directly on the graph. We normalize scores according to Appendix E. We use tasks listed in Appendix E.1. In the aggregate graph on the left, we average the performance of SAC [39], MH-SAC [121], BRC (single-task) and BRC (multi-task) between all continuous action benchmarks shown in Figure 1, summing to 171 tasks. For the BRO algorithm, we use the fast implementation with UTD = 2 according to Nauman et al. [77].

**Figure 2.** We show the performance of single-task BRC initialized from scratch, as well as single-task BRC initialized with the parameters of a pretrained multi-task BRC model. We use the task configurations described in Appendix E.2. To showcase the sample efficiency, we report the average performance in two time-steps denoted on the graph: in the final step of the training, as well as in the middle step of the training. We normalize the scores according to procedure described in Appendix D.

**Figure 3.** We report metrics from the final step of the 1M step training in the HumanoidBench-Medium benchmark with tasks listed in Appendix E.1. We use single and multi-task BRC algorithms. Besides regular BRC, we study two other variants: one using non-distributional Q-learning without reward normalization, and one using non-distributional Q-learning with reward normalization. In the left figure, we report the relative variance ($\sigma/\mu$) for temporal difference (TD) loss, gradient norm, and policy entropy conditioned on different tasks. In the right figure, we report the average normalized final performance of single and multi-task learners. We use normalization detailed in Appendix E and average over 5 random seeds for each method. We calculate 95% confidence interval using bootstrapping on random seeds [4].

**Figure 4.** We report final performance on the HumanoidBench-Medium benchmark detailed in Appendix E.1 after 1M environment steps per task. We consider single and multi-task BRC models with different critic architectures and varying width according to Table 5. In particular, we consider the vanilla MLP [39], SimBa [63], BroNet with MSE loss [77] and our proposed BroNet with cross-entropy loss. We calculate 95% confidence interval using bootstrapping on random seeds [4].

**Figure 5.** We consider single-task BRC, multi-task BRC and multi-task BRC with separate heads architecture [121, 61] on the HumanoidBench-Medium benchmark as described in Appendix E.1, trained for 1M steps. On the left Figure, we report the final performance for different widths of the critic model, as described in Table 5. In the right figure, we report conflicting gradients for the same models considered in the left figure. In particular, we report the percentage of conflicting gradients, with gradient conflict calculated according to the methodology presented in Yu et al. [120].

There, gradient conflicts are defined by the cosine similarity between gradients stemming from different samples. Specifically, a conflict occurs when the cosine similarity is negative, indicating opposing gradient directions. When calculating gradient conflicts in multi-task setup, we calculate conflicts between transitions sampled from different tasks, whereas, in the single-task setup, we divide the batch into two parts and calculate conflicts between these two subbatches. We calculate 95% confidence interval using bootstrapping on random seeds [4].

**Figure 6.** We report the final performance after 1M steps of training on HumanoidBench-Medium benchmark as described in Appendix E.1. We consider three design choices: scaled Q-value model denoted as $SQ$ (we consider using BroNet with cross-entropy loss using 4096 units of width, otherwise using 512 units of width); cross-entropy denoted as $CE$ (we consider the cross-entropy loss via categorical RL and reward normalization, otherwise we use non-distributional RL with the mean squared error loss); and learnable task embeddings denoted as $TE$ (we consider learning task embeddings by backpropagating the critic loss, otherwise we use the separate heads design). We evaluate all combinations of the described techniques, with $CE + TE + SQ$ being the base BRC model. We calculate 95% confidence interval using bootstrapping on random seeds [4]. In the right figure, we report exact Shapley values, which we calculate based on the runs presented in the left part of the figure.

**Figure 7.** We present the learned BRC task embeddings after 500k steps of online training on the MW+DMC benchmark. We take the 32-dimensional embeddings and extract first two principal components using the PCA algorithm.

**Figure 9.** We report the training curves for the HumanoidBench-Hard benchmark as described in Appendix E.1. We consider single and multi-task BRC, SimBaV2 [64], BRO-Fast [77] and TD-MPC2 [41]. All baseline algorithms use the configurations proposed in their respective manuscripts. The left figure shows environment steps per task, and the right figure shows the total gradient steps when learning all tasks. We calculate 95% confidence interval using bootstrapping on random seeds [4]. BRC uses 5 random seeds.

**Figure 10.** We report the final performance of BRC trained on different subsets of the HumanoidBench-Medium benchmark for 1M environment steps, given different critic widths as described in Table 5. We consider subsets of 3, 6, and 9 tasks from the benchmark: the 9 tasks variant uses all tasks listed in Appendix E.1 (i.e. walk, stand, run, stair, crawl, pole, slide, hurdle and maze); the 6 tasks variant trains on walk, stand, stair, crawl, pole and slide tasks and is consistent with the HumanoidBench-Medium transfer setup described in Appendix E.2; and the 3 tasks variant trains on walk, stair and slide tasks. In the left figure, we report the average final performance on the 3 tasks used in the training of all variants (i.e. walk, stair and slide tasks). In the right, we report the performance of the models trained on 6 and 3 tasks when transferred to 3 new tasks (run, hurdle and maze). On both graphs, we show the performance of single-task BRC for comparison and calculate 95% confidence interval using bootstrapping on 3-5 random seeds [4].

**Figure 11.** On the left Figure, we show the training curves averaged over 15 transfer tasks from HumanoidBench-Medium, HumanoidBench-Hard and MetaWorld listed in Appendix E.2. We report the first 500k steps of training, and show three algorithms: single-task BRC trained from scratch; single-task BRC initialized with the multi-task BRC trained on different tasks (results consistent with Figure 2 and description in Appendix E.2); and single-task BRC initialized with the replay buffer stemming from pretraining of the multi-task BRC model (basing on the setup proposed in Ball et al. [8]). In the right figure, we report the performance on 29 transfer tasks from the ShadowHand benchmark listed in Appendix E.2. In both graphs, we transfer 3 model seeds and calculate 95% confidence interval using bootstrapping on random seeds [4].

**Figure 12.** We focus on the MW+DMC benchmark. In the left, we report the final performance resulting from offline training according to Hansen et al. [41]. Additionally, we report the GPU days required to finish the training, assuming an 80 GB A100 graphics unit. On the right, we report the minimal amount of environment steps required to reach a fraction (50%, 75% and 100%) of the final multi-task TD-MPC2 performance on the MW+DMC benchmark [41], depending on the BRC model width. We additionally show a linear fit in the logarithmic space achieved using ordinary least squares regression. We run 5 seeds per model, except for the 1B for which we run 3 seeds.

## D.2 Baselines

**SAC.** Soft Actor-Critic [39] is an off-policy actor–critic method that maximizes the sum of expected returns and an entropy term, using twin Q-functions and a stochastic Gaussian policy. We use hyperparameters provided in Nauman et al. [77] and we rerun SAC for all environments except for HumanoidBench-Hard, where we use the results provided in Sferrazza et al. [93].

**MH-SAC.** Multi-Head SAC [120] shares a common encoder but attaches a separate head for each task, for both actor and critic. Thus the network produces task-specific outputs while retaining shared feature extraction. Similarly to BRC, we use the hyperparameters of single-task SAC [77].

**BRO.** The Bigger, Regularized, Optimistic [77] builds on SAC and scales the critic network by using the BroNet architecture. BroNet architecture (Figure 13) leverages layer normalization to restrict the complexity of the learned Q-value function. We use the low UTD variant of BRO (BRO-Fast), and use the results provided in the original manuscript. For missing environments, we run BRO-Fast using the original hyperparameters [77].

**SimBa.** SimBa architecture [63] is a follow-up work to BroNet, where authors propose to change the initial layer normalization to running statistics normalization. We run SimBa using the code from the official repository.

**SimBaV2.** SimBaV2 architecture [64] is a follow-up work to SimBa, where authors propose to change the layer normalization to hyperspherical normalization. We run SimBaV2 using the official repository using the proposed hyperparameters [64].

**TD-MPC2.** TD-MPC [41] learns a latent dynamics model and performs model-predictive control by back-propagating through imagined latent trajectories. We use the results provided in the original TD-MPC2 manuscript [41] and for HumanoidBench-Hard we use the results provided in Sferrazza et al. [93].

**DreamerV3.** DreamerV3 [40] trains a recurrent latent world model from pixels, imagines rollouts to update an entropy-regularized actor–critic. We use the results provided in TD-MPC2 manuscript [41].

**PCGrad.** PCGrad [120] is a multi-task method that resolves gradient interference in multi-task learning by projecting each task's gradient onto the orthogonal complement of any other task's conflicting component before the parameters are updated. We use the results provided in He et al. [44].

**PaCo.** Parameter-Compositional learning [99] is a multi-task method that factors each network's weights into a low-rank shared basis and task-specific composition vectors, so each task's parameters are expressed as a linear combination in the common subspace. We use the results provided in He et al. [44].

**CTPG.** Cross-Task Policy Guidance [44] is a multi-task method that trains an auxiliary selector that decides, at every step, which task-conditioned controller should act; the resulting trajectories provide data to update all controllers simultaneously. We use the results provided in He et al. [44].

**GAMT.** Geometry-Aware Multi-Task learning [49] is a multi-task method that feeds a frozen PointNet encoder that converts each object's point cloud into a shape embedding, then conditions a shared dexterous-hand policy on that embedding, allowing manipulation behaviors to generalize across object geometries. We use the results provided in Huang et al. [49].

**DDPG+HER.** Deep Deterministic Policy Gradient combined with Hindsight Experience Replay [5] relabels each stored transition with alternative goals that were actually achieved, permitting goal-conditioned value and policy learning from sparse binary rewards. We use the results provided in Huang et al. [49].

# E Considered Benchmarks

In this section, we describe the multi-task benchmarks that we considered in our studies, as well as the normalization scheme that we used in the evaluation. We use default episode lengths for all environments, except for MetaWorld, where we truncate after 200 steps following prior work [41, 77]. We do not use action repeat wrappers.

### E.1 Multi-task benchmarks

- **MetaWorld** (MW) - we use the full benchmark of 50 tasks [121]. The benchmark uses a relatively low observation and action dimensionality of $|S| = 39$ and $|A| = 4$.

    Training tasks: *assembly, basketball, bin-picking, box-close, button-press-topdown, button-press-topdown-wall, button-press, button-press-wall, coffee-button, coffee-pull, coffee-push, dial-turn, disassemble, door-close, door-lock, door-open, door-unlock, hand-insert, drawer-close, drawer-open, faucet-open, faucet-close, hammer, handle-press-side, handle-press, handle-pull-side, handle-pull, lever-pull, pick-place-wall, pick-out-of-hole, pick-place, plate-slide, plate-slide-side, plate-slide-back, plate-slide-back-side, peg-insert-side, peg-unplug-side, soccer, stick-push, stick-pull, push, push-wall, push-back, reach, reach-wall, shelf-place, sweep-into, sweep, window-open, window-close*

- **HumanoidBench-Medium** (HB-MEDIUM) - Here, we consider the benchmark of locomotion tasks without Shadow Hands [63]. We use all tasks with common embodiment with $|S| = 51$ and $|A| = 19$.

    Training tasks: *walk, stand, run, stair, crawl, pole, slide, hurdle, maze*

- **HumanoidBench-Hard** (HB-HARD) - Here, we consider the benchmark of locomotion and manipulation tasks with Shadow Hands [93]. We use 20 tasks with varying state dimensionality, leading $|S| = 307, |A| = 61$.

    Training tasks: *walk, stand, run, stair, crawl, pole, slide, hurdle, maze, sit-simple, sit-hard, balance-simple, balance-hard, reach, spoon, window, insert-small, insert-normal, bookshelf-simple, bookshelf-hard*

- **DeepMind Control Dogs** (DMC-HARD) - We use all dog tasks from the DMC benchmark [101] leading to 4 common embodiment tasks with $|S| = 223$ and $|A| = 38$.

    Training tasks: *dog-stand, dog-walk, dog-trot, dog-run*

- **DeepMind Control Humanoids** (DMC-HARD) - We use all humanoid tasks from the DMC benchmark [101] leading to 3 common embodiment tasks with $|S| = 67$ and $|A| = 24$.

    Training tasks: *humanoid-stand, humanoid-walk, humanoid-run*

- **ShadowHand** (SH) - We use all training tasks as proposed in Huang et al. [49], leading to 85 manipulation shapes with $|S| = 68$ and $|A| = 20$.

    Training tasks: *e-toy-airplane, knife, flat-screwdriver, elephant, apple, scissors, i-cups, cup, foam-brick, pudding-box, wristwatch, padlock, power-drill, binoculars, b-lego-duplo, ps-controller, mouse, hammer, f-lego-duplo, piggy-bank, can, extra-large-clamp, peach, a-lego-duplo, racquetball, tuna-fish-can, a-cups, pan, strawberry, d-toy-airplane, wood-block, small-marker, sugar-box, ball, torus, i-toy-airplane, chain, j-cups, c-toy-airplane, airplane, nine-hole-peg-test, water-bottle, c-cups, medium-clamp, large-marker, h-cups, b-colored-wood-blocks, j-lego-duplo, f-toy-airplane, toothbrush, tennis-ball, mug, sponge, k-lego-duplo, phillips-screwdriver, f-cups, c-lego-duplo, d-marbles, d-cups, camera, d-lego-duplo, golf-ball, k-toy-airplane, b-cups, softball, wine-glass, chips-can, cube, master-chef-can, alarm-clock, gelatin-box, h-lego-duplo, baseball, light-bulb, banana, rubber-duck, headphones, i-lego-duplo, b-toy-airplane, pitcher-base, j-toy-airplane, g-lego-duplo, cracker-box, orange, e-cups*

- **Diverse embodiment** (MW+DMC) - We use the diverse embodiment benchmark proposed in Hansen et al. [41]. This benchmark uses 80 tasks from the MetaWorld and DeepMind Control benchmarks, with a total of 12 different robot embodiments. The benchmark uses $|S| = 39$ and $|A| = 6$.

    Training tasks: *assembly, basketball, bin-picking, box-close, button-press-topdown, button-press-topdown-wall, button-press, button-press-wall, coffee-button, coffee-pull, coffee-push, dial-turn, disassemble, door-close, door-lock, door-open, door-unlock, hand-insert, drawer-close, drawer-open, faucet-open, faucet-close, hammer, handle-press-side, handle-press, handle-pull-side, handle-pull, lever-pull, pick-place-wall, pick-out-of-hole, pick-place, plate-slide, plate-slide-side, plate-slide-back, plate-slide-back-side, peg-insert-side, peg-unplug-side, soccer, stick-push, stick-pull, push, push-wall, push-back, reach, reach-wall, shelf-place, sweep-into, sweep, window-open, window-close, walker-stand, walker-walk, walker-run, cheetah-run, reacher-easy, reacher-hard, acrobot-swingup, pendulum-swingup, cartpole-balance, cartpole-balance-sparse, cartpole-swingup, cartpole-swingup-sparse, ball-in-cup-catch, finger-spin, finger-turn-easy,*

*finger-turn-hard, fish-swim, hopper-stand, hopper-hop, cheetah-run-backwards, cheetah-run-front, cheetah-run-back, cheetah-jump, walker-walk-backwards, walker-run-backwards, hopper-hop-backwards, reacher-three-easy, reacher-three-hard, ball-in-cup-spin, pendulum-spin*

- **Atari** (ATARI) - Here, we consider all 26 tasks from the Atari 100k benchmark [51]. We use full ALE actions space [10], leading to $|S| = 84 \times 84 \times 4$ and $|A| = 18$.

  Training tasks: *alien, amidar, assault, asterix, bankheist, battlezone, boxing, breakout, choppercommand, crazyclimber, demonattack, freeway, frostbite, gopher, hero, jamesbond, kangaroo, krull, kungfumaster, mspacman, pong, privateeye, qbert, roadrunner, seaquest, updown*

## E.2 Transfer learning benchmarks

Here, we detail the multi-task and transfer configurations used in our experiments. The difference with respect to previous subsection is that in these benchmarks we had to leave from multi-task training tasks for transfer evaluation.

- **MetaWorld** (MW) - for transfer, we left 5 hardest tasks according to Hansen et al. [41].

  Training tasks (45 tasks): *basketball, bin-picking, box-close, button-press-topdown, button-press-topdown-wall, button-press, button-press-wall, coffee-button, coffee-pull, coffee-push, door-close, door-lock, door-open, door-unlock, hand-insert, drawer-close, drawer-open, faucet-open, faucet-close, hammer, handle-press-side, handle-press, handle-pull-side, handle-pull, pick-place-wall, pick-out-of-hole, pick-place, plate-slide, plate-slide-side, plate-slide-back, plate-slide-back-side, peg-insert-side, peg-unplug-side, soccer, stick-push, stick-pull, push, push-wall, reach, reach-wall, shelf-place, sweep-into, sweep, window-open, window-close*
  Transfer tasks (5 tasks): *assembly, disassemble, dial-turn, lever-pull, push-back*

- **HumanoidBench-Medium** (HB-MEDIUM) - for transfer, we left 3 hardest tasks according to Lee et al. [63].

  Training tasks (6 tasks): *walk, stand, stair, crawl, pole, slide*
  Transfer tasks (3 tasks): *run, hurdle, maze*

- **HumanoidBench-Hard** (HB-HARD) - for transfer, we left 7 tasks categorized as "hard manipulation" in Sferrazza et al. [93].

  Training tasks (20 tasks): *walk, stand, run, stair, crawl, pole, slide, hurdle, maze, sit-simple, sit-hard, balance-simple, balance-hard, reach, spoon, window, insert-small, insert-normal, bookshelf-simple, bookshelf-hard*
  Transfer tasks (7 tasks): *truck, powerlift, room, door, basketball, push, cabinet*

- **ShadowHand** (SH) - we followed the training/transfer division proposed in Huang et al. [49].

  Training tasks (85 tasks): *e-toy-airplane, knife, flat-screwdriver, elephant, apple, scissors, i-cups, cup, foam-brick, pudding-box, wristwatch, padlock, power-drill, binoculars, b-lego-duplo, ps-controller, mouse, hammer, f-lego-duplo, piggy-bank, can, extra-large-clamp, peach, a-lego-duplo, racquetball, tuna-fish-can, a-cups, pan, strawberry, d-toy-airplane, wood-block, small-marker, sugar-box, ball, torus, i-toy-airplane, chain, j-cups, c-toy-airplane, airplane, nine-hole-peg-test, water-bottle, c-cups, medium-clamp, large-marker, h-cups, b-colored-wood-blocks, j-lego-duplo, f-toy-airplane, toothbrush, tennis-ball, mug, sponge, k-lego-duplo, phillips-screwdriver, f-cups, c-lego-duplo, d-marbles, d-cups, camera, d-lego-duplo, golf-ball, k-toy-airplane, b-cups, softball, wine-glass, chips-can, cube, master-chef-can, alarm-clock, gelatin-box, h-lego-duplo, baseball, light-bulb, banana, rubber-duck, headphones, i-lego-duplo, b-toy-airplane, pitcher-base, j-toy-airplane, g-lego-duplo, cracker-box, orange, e-cups*
  Transfer tasks (29 tasks): *rubiks-cube, dice, bleach-cleanser, pear, e-lego-duplo, pyramid, stapler, flashlight, large-clamp, a-toy-airplane, tomato-soup-can, fork, cell-phone, m-lego-duplo, toothpaste, flute, stanford-bunny, a-marbles, potted-meat-can, timer, lemon, utah-teapot, train, g-cups, l-lego-duplo, bowl, door-knob, mustard-bottle, plum*

- **Atari** (ATARI) - we used the tasks from Atari100k [51] for multi-task training.

  Training tasks (26 tasks): *alien, amidar, assault, asterix, bankheist, battlezone, boxing, breakout, choppercommand, crazyclimber, demonattack, freeway, frostbite, gopher, hero, jamesbond, kangaroo, krull, kungfumaster, mspacman, pong, privateeye, qbert, roadrunner, seaquest, updown*

Transfer tasks (20 tasks): *asteroids, atlantis, beamrider, berzerk, bowling, centipede, double-dunk, fishingderby, gravitar, icehockey, namethisgame, pitfall, riverraid, stargunner, timepilot, tutankham, venture, videopinball, wizardofwor, yarsrevenge*

## E.3 Score normalization

We normalize the scores according to practices from previous works, with the goal of bounding the returns between 0 and 1. In MetaWorld, we just report unnormalized success rates (which per definition or bounded). In HumanoidBench-Medium and HumanoidBench-Hard, we report returns, which we normalize according to Equation 4 and scores sourced from Sferrazza et al. [93]. In DMC, we report returns, which we normalize by diving by 1000 (according to Nauman et al. [77]). In ShadowHand, we report success rates. Finally, in Atari tasks, we report human normalized returns [72] according to Equation 4 and values presented in Agarwal et al. [4].

$$Normalized\ Returns = \frac{Returns - Random\ Returns}{Optimal\ Returns - Random\ Returns} \tag{4}$$

For Atari, we use the scores provided in Agarwal et al. [4]. The table below details the normalization scores used in the HumanoidBench benchmark.

Table 2: Random and success scores for HumanoidBench tasks

| Task | Random Score | Optimal Score |
|------|--------------|---------------|
| h1-crawl-v0 | 272.658 | 700.0 |
| h1-hurdle-v0 | 2.214 | 700.0 |
| h1-maze-v0 | 106.441 | 1200.0 |
| h1-pole-v0 | 20.090 | 700.0 |
| h1-run-v0 | 2.020 | 700.0 |
| h1-slide-v0 | 3.191 | 700.0 |
| h1-stair-v0 | 3.112 | 700.0 |
| h1-stand-v0 | 10.545 | 800.0 |
| h1-walk-v0 | 2.377 | 700.0 |
| h1hand-balance_hard-v0 | 10.032 | 800.0 |
| h1hand-balance_simple-v0 | 10.170 | 800.0 |
| h1hand-basketball-v0 | 8.979 | 1200.0 |
| h1hand-bookshelf_hard-v0 | 14.848 | 2000.0 |
| h1hand-bookshelf_simple-v0 | 16.777 | 2000.0 |
| h1hand-crawl-v0 | 278.868 | 800.0 |
| h1hand-door-v0 | 2.771 | 600.0 |
| h1hand-hurdle-v0 | 2.371 | 700.0 |
| h1hand-insert_normal-v0 | 1.673 | 350.0 |
| h1hand-insert_small-v0 | 1.653 | 350.0 |
| h1hand-maze-v0 | 106.233 | 1200.0 |
| h1hand-package-v0 | -10040.932 | 1500.0 |
| h1hand-pole-v0 | 19.721 | 700.0 |
| h1hand-powerlift-v0 | 17.638 | 800.0 |
| h1hand-push-v0 | -526.800 | 700.0 |
| h1hand-reach-v0 | -50.024 | 12000.0 |
| h1hand-room-v0 | 3.018 | 400.0 |
| h1hand-run-v0 | 1.927 | 700.0 |
| h1hand-sit_hard-v0 | 2.477 | 750.0 |
| h1hand-sit_simple-v0 | 10.768 | 750.0 |
| h1hand-slide-v0 | 3.142 | 700.0 |
| h1hand-spoon-v0 | 4.661 | 650.0 |
| h1hand-stair-v0 | 3.161 | 700.0 |
| h1hand-stand-v0 | 11.973 | 800.0 |
| h1hand-truck-v0 | 562.419 | 3000.0 |
| h1hand-walk-v0 | 2.505 | 700.0 |
| h1hand-window-v0 | 2.713 | 650.0 |

## F  Hyperparameters

We detail the hyperparameters used in our experiments in Tables 3 and 4 below. As discussed in Section 4, we use a single hyperparameter configuration across all tested tasks, showcasing robustness of our approach. We release the code at the following link: `https://github.com/naumix/BiggerRegularizedCategorical`.

Table 3: Hyperparameters for SAC+BRC

| Hyperparameter | Value |
|---|---|
| UTD | 2 |
| Action repeat | 1 |
| Embedding size | 32 |
| Discount rate | 0.99 |
| Optimizer | AdamW |
| Num atoms | 101 |
| $V_{min}$ | $-10$ |
| $V_{max}$ | 10 |
| Polyak $\tau$ | 5e-3 |
| Target update frequency | 1 |
| Weight decay | 1e-4 |
| Batch size | 1024 |
| Buffer size per task | 1e6 |
| Actor learning rate | 3e-4 |
| Critic learning rate | 3e-4 |
| Temperature learning rate | 3e-4 |
| Initial temperature | 0.1 |
| Target entropy | $|\mathcal{A}|/2$ |
| Actor architecture | BroNet |
| Actor depth | 1 |
| Actor width | 256 |
| Critic architecture | BroNet |
| Critic depth | 2 |
| Critic width | 4096 |
| Num critics | 2 |

Table 4: Hyperparameters for DrQ+BRC

| Hyperparameter | Value |
|---|---|
| UTD | 2 |
| Action repeat | 1 |
| Embedding size | 32 |
| Discount rate | 0.99 |
| Optimizer | AdamW |
| Num atoms | 101 |
| $V_{min}$ | $-10$ |
| $V_{max}$ | 10 |
| Polyak $\tau$ | 5e-3 |
| Target update frequency | 1 |
| Weight decay | 1e-4 |
| Batch size | 256 |
| Buffer size per task | 1e5 |
| Learning rate | 1e-4 |
| Encoder | Impala |
| Encoder channels | (128, 256, 256) |
| Pre-critic norm | Layer norm |
| Critic | Dense |
| Critic depth | 2 |
| Critic width | 512 |
| Data augmentation | True |
| Start $\epsilon$ | 1.0 |
| End $\epsilon$ | 0.01 |
| $\epsilon$ decay steps | 5000 |
| N-step | 3 |

## G  Additional Efficiency Results

We include wall-clock measurements comparing efficiency of BRO [77] and multi-task BRC across four model sizes for HumanoidBench (HB-Medium benchmark). We generate these wall-clock estimates using an uniform setup using a single H100 GPU with 16 CPU cores of AMD EPYC 7742 processor. The results, summarized in the table below, confirm that multi-task learning via BRC offers significant gains in efficiency, even at reduced model scales.

Table 5: Multi-task learning via BRC offers wall-clock improvements, even when compared to BRO which uses 8x smaller batch size and 64x smaller critic.

| | BRO-4M | BRC-4M | BRC-16M | BRC-64M | BRC-256M |
|---|---|---|---|---|---|
| Final Performance | 0.57 | 0.65 | 0.69 | 0.83 | 0.95 |
| Minutes to perform 1M steps (all tasks) | 1013 | 242 | 256 | 336 | 658 |
| Minutes to reach BRO final performance | 1013 | 157 | 138 | 154 | 263 |
| Wall-clock improvement | 1 | 6.4 | 7.3 | 6.6 | 3.9 |

## H  Training Curves

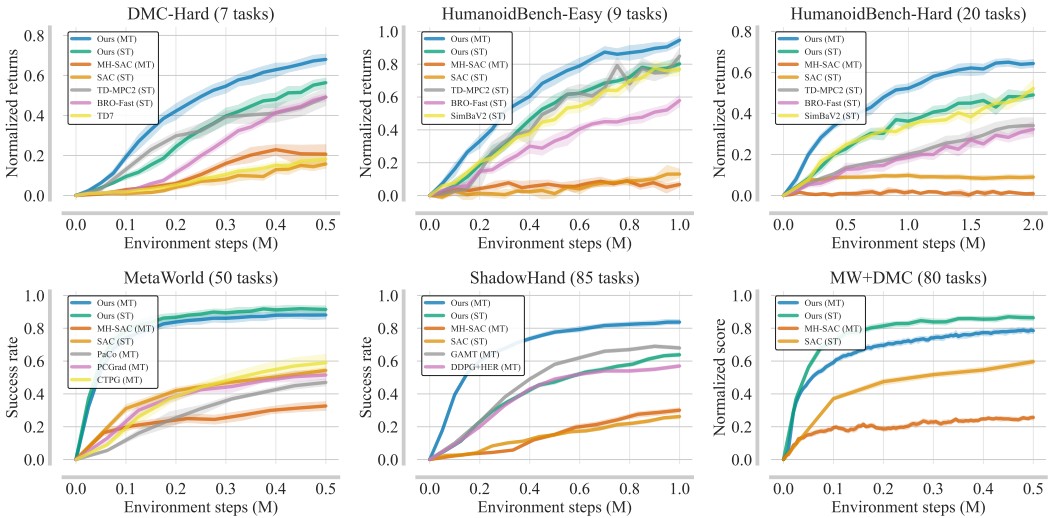

Figure 16: **Aggregate training curves**. Y-axis denotes the performance metric, and X-axis denotes environment steps. We report 95% confidence interval calculated via bootstrapping.

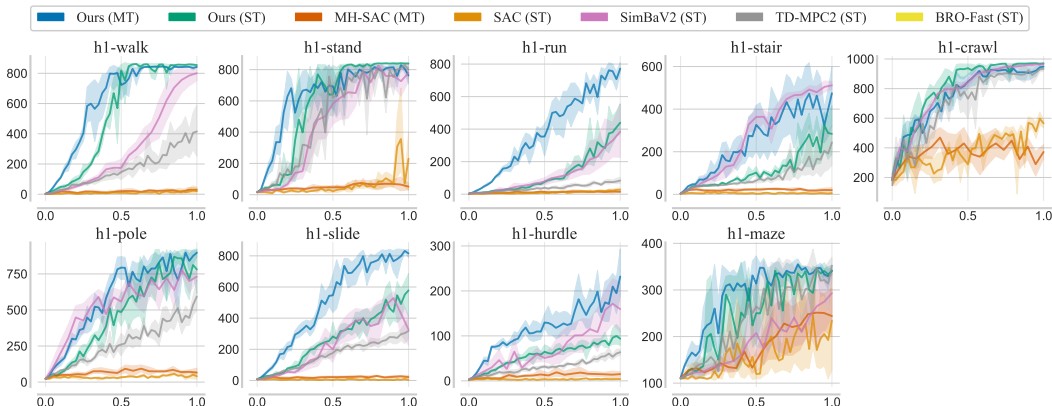

Figure 17: **Unnormalized training curves for the HumanoidBench-Medium benchmark**. Y-axis denotes sum of episodic returns and X-axis denotes environment steps. We report 95% confidence interval calculated via bootstrapping.

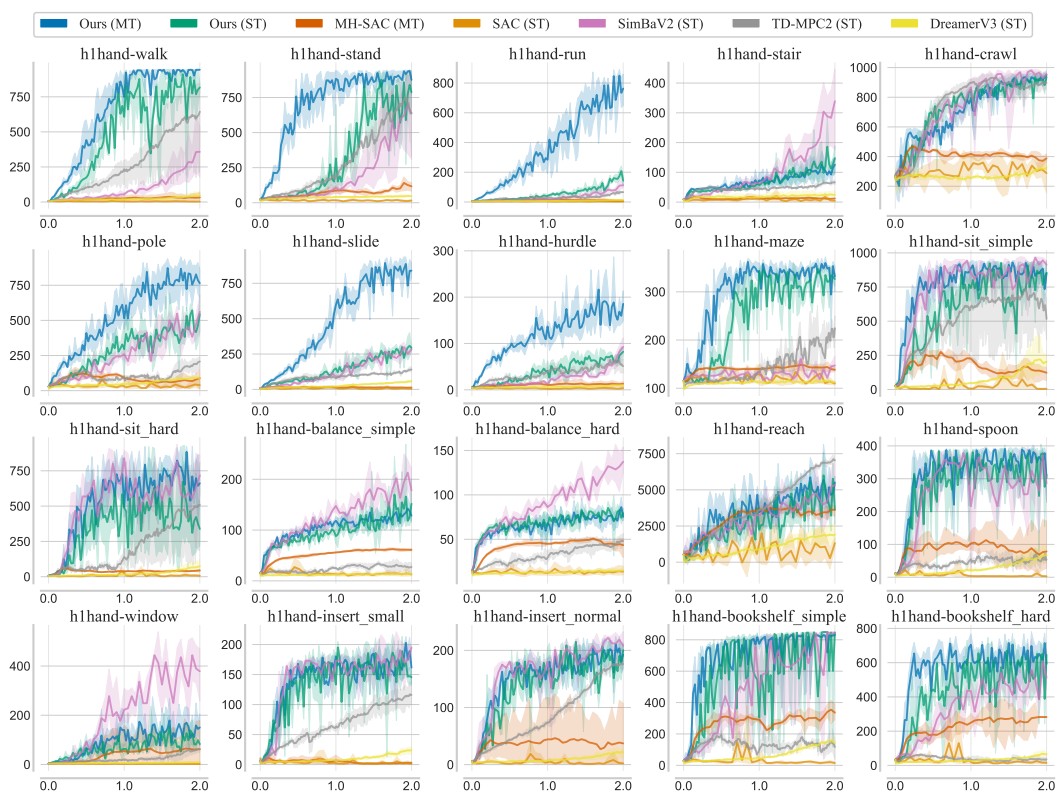

Figure 18: **Unnormalized training curves for the HumanoidBench-Hard benchmark**. Y-axis denotes sum of episodic returns and X-axis denotes environment steps. We report 95% confidence interval calculated via bootstrapping.

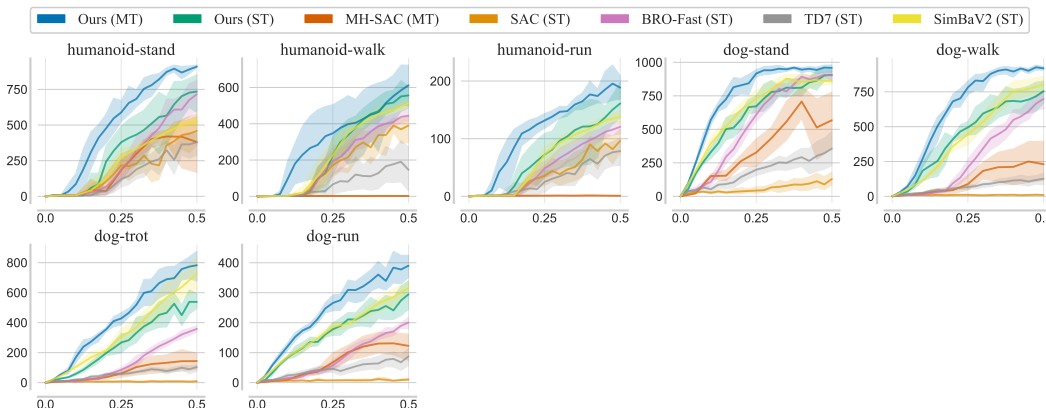

Figure 19: **Unnormalized training curves for the DMC-Hard benchmark**. Y-axis denotes sum of episodic returns and X-axis denotes environment steps. We report 95% confidence interval calculated via bootstrapping.

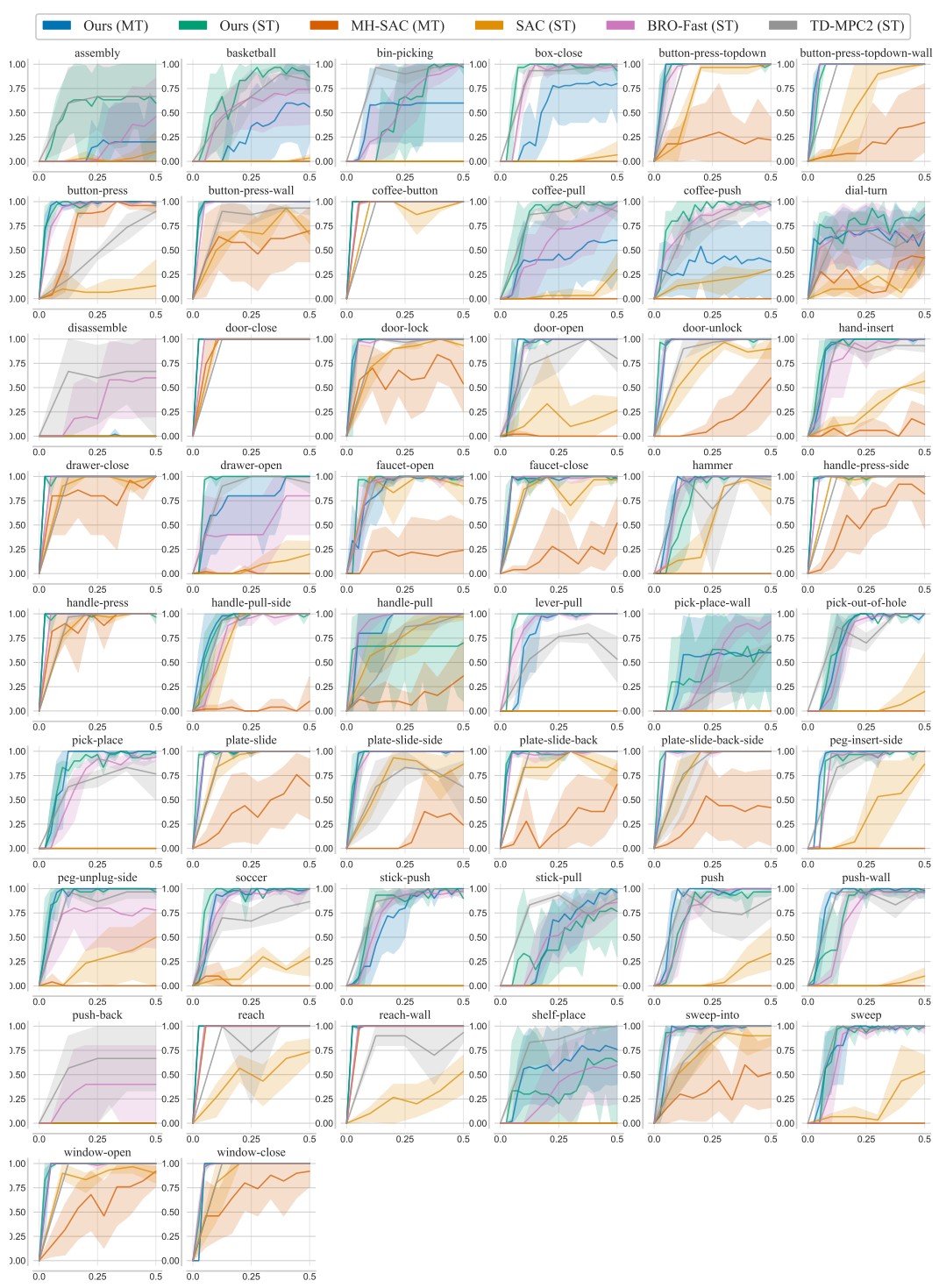

Figure 20: **Unnormalized training curves for the MetaWorld benchmark**. Y-axis denotes the success rate and X-axis denotes environment steps. We report 95% confidence interval calculated via bootstrapping.

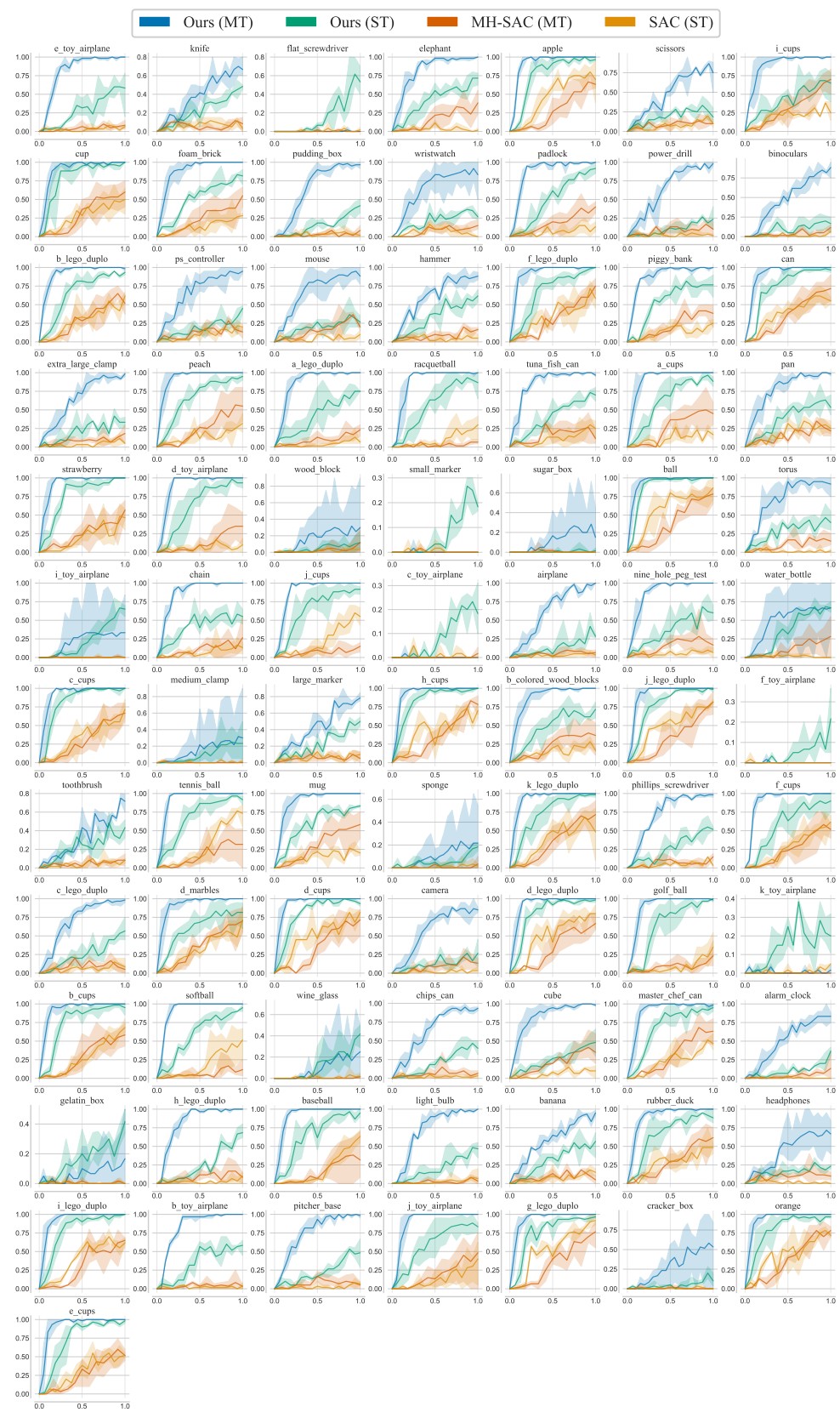

Figure 21: **Unnormalized training curves for the ShadowHand benchmark**. Y-axis denotes the success rate and X-axis denotes environment steps. We report 95% confidence interval calculated via bootstrapping.

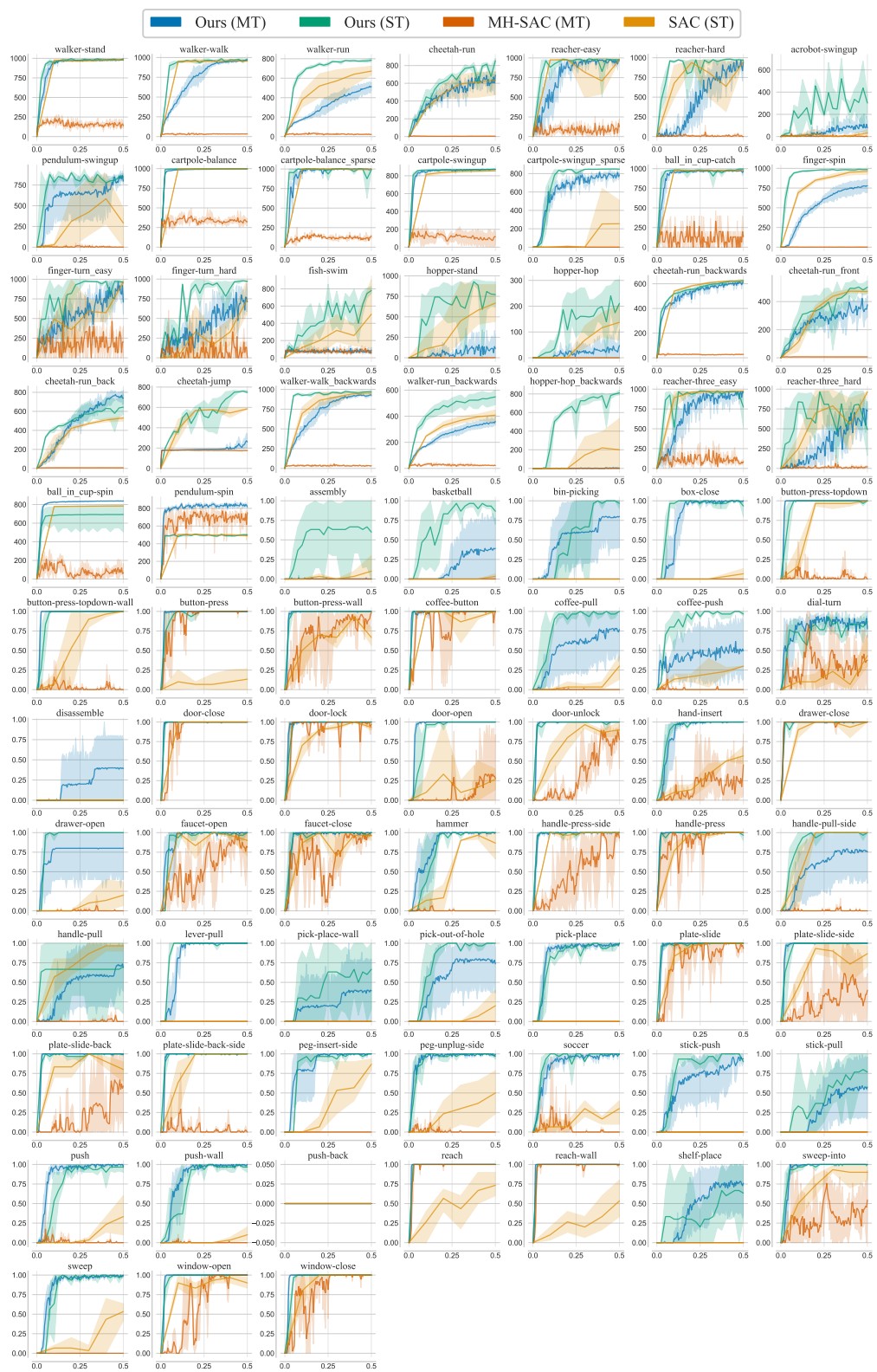

Figure 22: **Unnormalized training curves for the MW+DMC benchmark**. For MetaWorld tasks, Y-axis denotes the success rate whereas for DMC tasks it denotes sum of episodic returns. X-axis denotes environment steps. We report 95% confidence interval calculated via bootstrapping.

Table 6: Atari-100k benchmark scores.

| Game | Human | SimPLe | DER | SPR | DrQ | Ours (ST) | Ours (MT) |
|---|---|---|---|---|---|---|---|
| Alien | 7127.7 | 616.9 | 739.9 | 841.9 | 865.2 | 1005.8 | 906.6 |
| Amidar | 1719.5 | 88.0 | 188.6 | 179.7 | 137.8 | 192.1 | 147.8 |
| Assault | 742.0 | 527.2 | 431.2 | 565.6 | 579.6 | 629.0 | 503.4 |
| Asterix | 8503.3 | 1128.3 | 470.8 | 962.5 | 763.6 | 1442.2 | 1285.4 |
| BankHeist | 753.1 | 34.2 | 51.0 | 345.4 | 232.9 | 959.7 | 104.5 |
| BattleZone | 37187.5 | 5184.4 | 10124.6 | 14834.1 | 10165.3 | 14905.0 | 20316.0 |
| Boxing | 12.1 | 9.1 | 0.2 | 35.7 | 9.0 | -0.1 | 45.1 |
| Breakout | 30.5 | 16.4 | 1.9 | 19.6 | 19.8 | 24.9 | 20.6 |
| ChopperCommand | 7387.8 | 1246.9 | 861.8 | 946.3 | 844.6 | 1867.5 | 1848.4 |
| CrazyClimber | 35829.4 | 62583.6 | 16185.3 | 36700.5 | 21539.0 | 79276.5 | 94606.8 |
| DemonAttack | 1971.0 | 208.1 | 508.0 | 517.6 | 1321.5 | 503.2 | 479.6 |
| Freeway | 29.6 | 20.3 | 27.9 | 19.3 | 20.3 | 31.6 | 28.4 |
| Frostbite | 4334.7 | 254.7 | 866.8 | 1170.7 | 1014.2 | 1413.9 | 715.7 |
| Gopher | 2412.5 | 771.0 | 349.5 | 660.6 | 621.6 | 512.0 | 1477.6 |
| Hero | 30826.4 | 2656.6 | 6857.0 | 5858.6 | 4167.9 | 10228.0 | 7472.3 |
| Jamesbond | 302.8 | 125.3 | 301.6 | 366.5 | 349.1 | 249.0 | 288.6 |
| Kangaroo | 3035.0 | 323.1 | 779.3 | 3617.4 | 1088.4 | 3644.0 | 1184.0 |
| Krull | 2665.5 | 4539.9 | 2851.5 | 3681.6 | 4402.1 | 5277.8 | 6764.8 |
| KungFuMaster | 22736.3 | 17257.2 | 14346.1 | 14783.2 | 11467.4 | 6310.5 | 13723.2 |
| MsPacman | 6951.6 | 1480.0 | 1204.1 | 1318.4 | 1218.1 | 6310.5 | 1682.1 |
| Pong | 14.6 | 12.8 | -19.3 | -5.4 | -9.1 | 1.9 | 0.2 |
| PrivateEye | 69571.3 | 58.3 | 97.8 | 86.0 | 3.5 | 42.7 | 99.8 |
| Qbert | 13455.0 | 1288.8 | 1152.9 | 866.3 | 1810.7 | 4781.2 | 1954.6 |
| RoadRunner | 7845.0 | 5640.6 | 9600.0 | 12213.1 | 11211.4 | 11887.0 | 18021.2 |
| Seaquest | 42054.7 | 683.3 | 354.1 | 558.1 | 352.3 | 434.6 | 635.2 |
| UpNDown | 11693.2 | 3350.3 | 2877.4 | 10859.2 | 4324.5 | 5290.3 | 7102.2 |
| **Mean** | 1.0 | 0.443 | 0.285 | 0.616 | 0.465 | 0.667 | 0.849 |

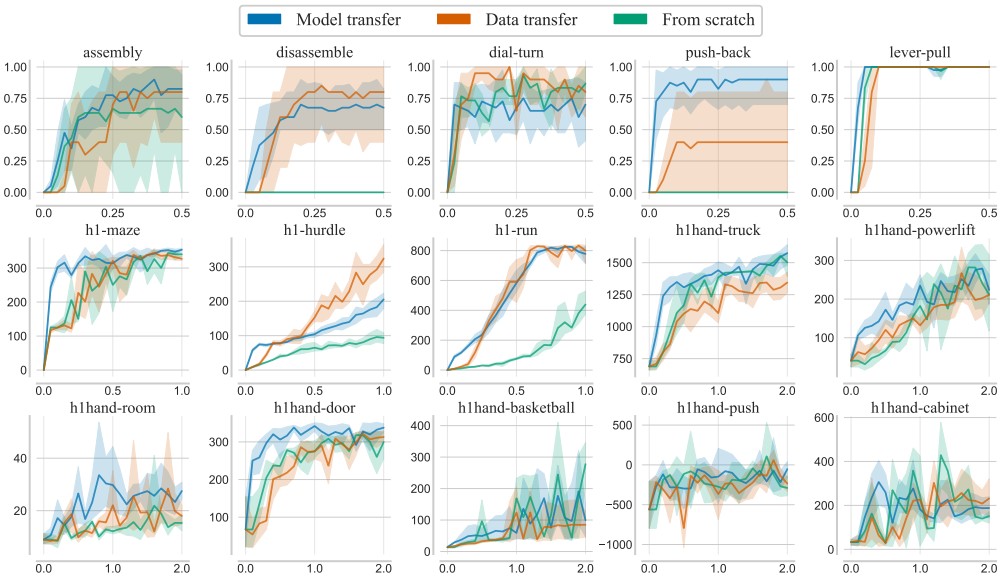

Figure 23: **Unnormalized training curves for the transfer tasks**. For MetaWorld tasks, Y-axis denotes the success rate whereas for other tasks it denotes sum of episodic returns. X-axis denotes environment steps. We report 95% confidence interval calculated via bootstrapping.

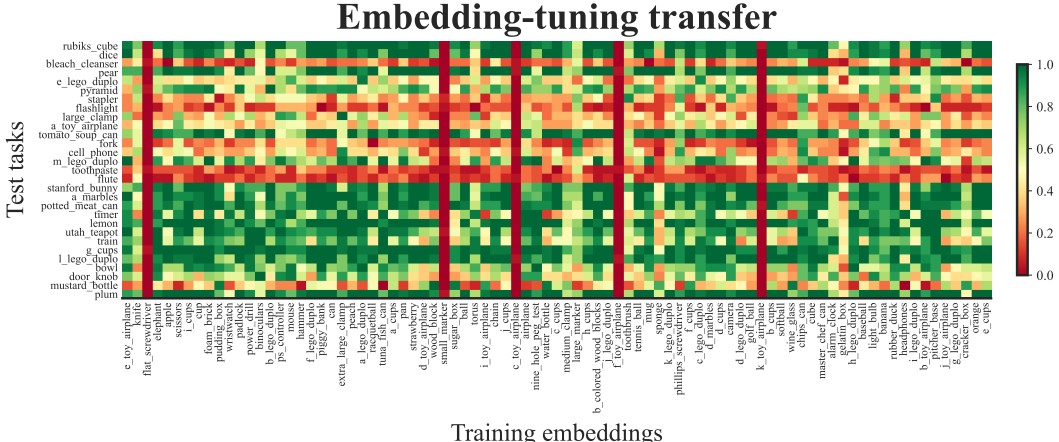

Figure 24: **Effectiveness of pretrained embeddings**. We show the performance of using pretrained embeddings to manipulate new shapes. Interestingly, most of the testing shapes can be manipulated via majority of the pretrained embeddings, showcasing that the individual policies do not overfit to the particular task.

