# OpenReview forum: "Bigger, Regularized, Categorical: High-Capacity Value Functions are Efficient Multi-Task Learners"
_NeurIPS.cc/2025/Conference — NeurIPS 2025 poster_

### Official Review · Reviewer_zxeb · 2025-06-27

**Clarity:** 3
**Significance:** 3
**Originality:** 2
**Rating:** 5
**Confidence:** 3

**Summary:**

This paper investigates three existing techniques in reinforcement learning (RL) to improve the capacity and learning of the value function in multi-task reinforcement learning. Specifically, the three techniques are distribution RL cross-entropy loss, larger ResNet-based critic network with LayerNorm, and learned task embeddings. The paper presents extensive experimental results on over 280 tasks across five benchmarks, providing insights for how improving and scaling critic learning can boost performance and generalization in the multi-task setting.

**Questions:**

Below questions might impact the evaluation:
1. Figure 11 (left) suggests that model transfer is more efficient than data transfer. However, this appears to contradict findings in [1] and the BRO paper, where data transfer is shown to be useful in single-task settings, particularly concerning network resets. Could you elaborate on this apparent discrepancy?
2. Is SAC+BRC the same as BRO-SR+CE (where SR represents Scaled-by-Resetting from [1] and CE is the cross entropy loss used in distribution RL)?
3. Are task IDs provided to the agent? This detail, along with a clearer definition of the learning problems, appears to be missing or not explicitly stated in the main text. Clarifying these aspects is crucial for a complete understanding of the experimental setup.
4. Error bars appear to be missing from several plots, including Figures 1, 2, and 3 (left). Could you please add these to represent the variability and statistical significance of the results?

Below are suggestions for improving the paper:
1. It is crucial to explain statistical significance in DRL experiments. I’d suggest at least mention Appendix D.1 in the caption of Figure 1.
2. In Figure 5 left, The comparison between "separate heads" and "single-task" isn't statistically significant. Their confidence intervals overlap, and their rank order flips at various points. To strengthen this claim, consider increasing the number of runs or adding more data points (e.g., testing additional network sizes).

[1] D'Oro, P., Schwarzer, M., Nikishin, E., Bacon, P. L., Bellemare, M. G., & Courville, A. Sample-Efficient Reinforcement Learning by Breaking the Replay Ratio Barrier. In The Eleventh International Conference on Learning Representations.

**Ethical Concerns:**

["NO or VERY MINOR ethics concerns only"]

**Final Justification:**

During the rebuttal, the authors' response addressed my questions regarding the distinction between model transfer and data transfer in this paper, as well as some algorithmic details. I have also read other reviews and the author's responses. While there are directions mentioned by other reviewers to further improve the paper, I think the current contributions of the paper are strong and of interest to the community. I maintain my positive assessment and support the acceptance of this paper.

**Limitations:**

Yes.

**Paper Formatting Concerns:**

No.

**Quality:**

4

**Strengths And Weaknesses:**

The paper's primary strength is its comprehensive and rigorous empirical investigation:
- It conducts experiments on a diverse set of benchmarks, including continuous and discrete control, with coverage of both dense and sparse reward settings.
- It provides further insights into performance improvement through a variety of analyses, including variance analysis of different variables, transferability to new tasks, the parameter scaling effect, and ablation studies.
- The paper is clearly written and relevant works are properly cited, which is helpful considering the extensive results and literature.

A potential weakness of the paper is its limited novelty. The proposed approach is a combination of well-established techniques, and while their application to the multi-task setting is new, the resulting performance improvements might be considered predictable by researchers familiar with these methods. The paper's main contribution is therefore an empirical one, demonstrating the effectiveness of this combination, rather than a conceptual one. Nevertheless, this is a valuable and well-executed study that will be of interest to the community.

---

> ### Author Rebuttal · Authors · 2025-07-30
>
> We thank the reviewer for the helpful review and suggestions. We are glad that the reviewer appreciates the comprehensiveness of our empirical results. To the best of our knowledge, this constitutes one of the broadest evaluations in value-based RL to date. Following reviewer’s suggestions, we made a number of changes to our manuscript that we discuss in detail below:
>
> > The proposed approach is a combination of well-established techniques…
>
> We appreciate the reviewer’s thoughtful summary. We agree that the individual components used in our approach are well-established in the RL and supervised learning literature. However, their systematic combination in the context of multi-task value-based RL has not been explored in prior work, and we believe this empirical contribution is both non-trivial and timely. In particular, many recent efforts in multi-task RL have relied on specialized architectural tricks or task-specific strategies. The fact that a relatively simple combination of scalable design principles can match or exceed specialist performance across various tasks and allow for efficient transfer, is in our view a surprising and valuable result. We hope this simplicity and robustness will make BRC a useful baseline for future research.
>
> > Figure 11 (left) suggests that model transfer is more efficient than data transfer…
>
> We thank the reviewer for this insightful comment. The key difference lies in the source of the offline data: our “data transfer” method initializes the replay buffer using data from different tasks than the target, whereas approaches like RLPD or network resets (as in the BRO paper) rely on offline data collected from the same target task. This difference in data relevance likely explains the performance discrepancy. As expected, initializing with high-quality target task data (as in RLPD/resets) can be more effective for accelerating learning than using cross-task data. However, a strength of our “data transfer” approach is that it does not require any data from the target task, making it applicable in settings where such data is unavailable or expensive to collect. To better highlight this difference, **we add the above argument to Section 5**. We also agree that the observed performance gap between data and model transfer in Figure 11 (left) is intriguing and highlights a promising direction for future research into cross-task transfer dynamics.
>
> > Is SAC+BRC the same as BRO-SR+CE?
>
> No, SAC+BRC is not equivalent to BRO-SR+CE, although there are overlapping components. Our approach (SAC+BRC) shares some design elements with BRO, such as using BroNet for architectural structure and distributional RL (we use categorical cross-entropy instead of quantiles). However, SAC+BRC does not include several BRO components, including: Scaled-by-Resetting (SR), Optimistic exploration and Quantile-based value heads. Instead, we use task embeddings and categorical values (CE loss), which we found to be both simpler and more scalable across tasks. For example, unlike BRO which reduces batch size for stability, our method scales well with large batch sizes needed in diverse multi-task settings. Our design goal was to retain components with high practical impact while simplifying the overall method. As a result, SAC+BRC can be seen as a simplified yet more performant variant of BRO, that also works well in multi-task settings. To make these differences clearer for the reader, **we add a paragraph to the Appendix in which we discuss the differences and similarities between BRC and BRO**.
>
> > Are task IDs provided to the agent?
>
> We thank the reviewer for highlighting this. Yes, task identifiers are provided to the agent. Specifically, we assign an integer ID to each task (from 1 to NumTasks), and use this ID to retrieve a corresponding task embedding vector. This embedding is concatenated to the observation and fed into both the actor and critic networks. As described in Section 3, the task embeddings are learned by backpropagating the critic loss. To improve clarity, **we have now added a paragraph in Section 3 explicitly detailing this mechanism, and we include a diagram in the appendix to visually illustrate the process**.
>
> > Error bars appear to be missing from several plots / link to Appendix D...
>
> We thank the reviewer for this suggestion. **We will add error bars to the listed figures in the camera-ready version, as well as link to Appendix D in Figure 1**.
>
> > The comparison between "separate heads" and "single-task" isn't statistically significant…
>
> We thank the reviewer for this observation. You are right that the performance of "separate heads" and "single-task" in Figure 5 (left) is not statistically different for models smaller than 250M based on the current confidence intervals. That said, we find this result consistent with trends observed in prior work - when model capacity is limited, "separate heads" often underperforms compared to single-task specialists. Our intent in Figure 5 was to explore whether increased model capacity mitigates this gap and the results suggest that "separate heads" learners can surpass single-task performance at larger model scales. We agree that the analysis would be stronger with more random seeds - **we are happy to add these in the camera-ready version**.
>
> We thank the reviewer again for their thoughtful review. We sincerely think that incorporating these changes will improve the reader’s experience and the impact of our work. Please inform us if the rebuttal answered reviewers' hesitations.

---

> > ### Comment · Reviewer_zxeb · 2025-08-02
> >
> > Thank you for the rebuttal. I appreciate the authors’ clarification on the difference between model transfer and data transfer, as well as some algorithmic details. I will keep my current positive assessment and advocate for acceptance of this work.

---

> ### Author Response · Authors · 2025-08-02
> **Thank you for the rebuttal**
>
> Thank you for your help in improving our work. We are happy to answer any further questions and ready to discuss potential suggestions.

---

### Official Review · Reviewer_9y5s · 2025-06-29

**Clarity:** 4
**Significance:** 2
**Originality:** 2
**Rating:** 5
**Confidence:** 4

**Summary:**

The paper proposes a new state-of-the-art method on multi task RL, spanning across continuous/discrete action space and state/vision-based input space. Their main sources of improvement were (1) using a well-normalized model (BRO), (2) parameter scaling the model, (3) employing cross-entropy loss, and (4) learnable task embedding. The normalized model allowed its size to be scaled up to 256x (Figure 4), while the cross-entropy loss effectively unified the scale of the loss between tasks and thus their gradient magnitude.

The authors extensively verify their method's effective in 7 multi-task benchmarks (DMC, HB, Metaworld, Atari, ...), showing that (1) the proposed method performs on-par with previous single task SOTA models and (2) surpasses those methods in multi-task setting. This leads to significant training speedup, taking 2x less samples and 40x less gradient steps to achieve the same performance as prior works.

Experiments were also made in pretrain-then-finetune regime (transfer learning), where they consider three types of transfer: full-finetuning (model transfer), tuning embedding only, and creating expert transitions (data transfer), and show that they outperform from-scratch models as well as prior works on transfer learning.

**Questions:**

1. I think the compute efficiency in Figure 9 (right) is not fairly compared to prior works. To my understanding each gradient step would take significantly different compute to perform, depending on batch size, buffer size, parameter size, etc. In this case I think the x-axis should be either FLOPs or wall-clock time.

2. The data transfer method sound more like JSRL [1] (or WSRL [2] but that's already cited) than RLPD.

3. In Figure 10 (right), juding by the closing gap between 3 task and 6 task as parameter scales, it seems like parameter scaling and task (data) scaling are not complementary, which is surprising. What do you think of this result? In that sense, I am curious of the results with 9 tasks as well.

4. Also just curious: in Figure 11 (left), how would embedding-tuning transfer perform compared to other transfer methods?

[1] Jump-Start Reinforcement Learning., Uchendu et al.
[2] Efficient Online Reinforcement Learning Fine-Tuning Need Not Retain Offline Data., Zhou et al.

**Ethical Concerns:**

["NO or VERY MINOR ethics concerns only"]

**Final Justification:**

The paper makes great contribution to the RL community by show that multi-task RL has superior scaling properties than when learning from individual tasks, and provides a simple method (learnable task embedding) to avoid gradient interference between tasks. Scaling the model size in RL has been one of the major challenges recently, and I think this paper is a meaningful step in this direction.

**Limitations:**

Yes

**Quality:**

4

**Strengths And Weaknesses:**

Strength
- Extensive experiment clearly shows the effectiveness of the method
- They provide sufficient analysis that justify each component well, as well as exhaustive ablation of their 3 components.
- The writing is clear and straightforward, easy to understand.

Weakness
- The method likely suffers from heavy memory constraint, due to large replay buffer (1M per task * num tasks) and large batch size (1024).
- The components are relatively common in the RL field, although I don't think this as a huge weakness.

---

> ### Author Rebuttal · Authors · 2025-07-30
>
> We thank the reviewer for their help in improving our manuscript. We are glad that the reviewer finds our experiments convincing. Following up on reviewer’s suggestions, we made the following improvements to our work:
>
> 1. We include new results comparing the wall-clock training time of our method using different model sizes and BRO on the HumanoidBench tasks. These results demonstrate that multi-task BRC significantly reduces training time while improving the final performance.
> 2. We have made a number of textual and structural improvements to the manuscript based on reviewer feedback. These are detailed in our point-by-point responses.
>
> Please find our responses below:
>
> > The method likely suffers from heavy memory constraint
>
> We agree that memory usage from the replay buffer is a limitation of our method. However, in practice, we find the memory footprint to be manageable in most settings. Assuming Float32, on the HB-Medium benchmark with 9 tasks, the full replay buffer (1M transitions per task) occupies only 4.40 GB. On MW with 50 tasks, the total memory usage reaches 16.65 GB. Such requirements are often within the capacity of standard research or cloud hardware. In scenarios with a significantly larger number of tasks, the memory cost would scale linearly, but practical workarounds exist. For instance, a lazy loading or prioritized task sampling strategy could reduce the need to keep the full buffer for all tasks in memory at once. **We now mention this in the limitations section** to make this trade-off more clear.
>
> > The components are relatively common in the RL field, although I don't think this as a huge weakness.
>
> We agree that the components we use are individually well-established in the RL literature. Following the bitter lesson we chose simple, scalable components to maximize performance while minimizing complexity. We think that the novelty of our work lies in how these elements are combined and in the new perspective we offer on their interaction in the context of stabilizing multi-task RL. Our empirical results (e.g. Fig 6) demonstrate that their combination leads to synergistic effects not observed when using each component in isolation.
>
> > I think the compute efficiency in Figure 9...
>
> We thank the reviewer for this suggestion. We agree that gradient steps alone do not fully capture compute cost, since they can vary significantly depending on batch size, model architecture, and hardware utilization. In response to this, **we have added a supplement that includes wall-clock time measurements**, along with a discussion on the trade-offs between different compute metrics (gradient steps, FLOPs, and wall-clock time). Our initial proposition focuses on gradient steps because while FLOPs are a useful theoretical measure, they do not account for practical factors like GPU parallelism and memory access patterns. Similarly, wall-clock time can vary substantially based on implementation details and hardware setup. That said, to better illustrate the practical efficiency of multi-task learning, we now include wall-clock measurements comparing wall-clock times for BRO and multi-task BRC variants across four model sizes for HumanoidBench (HB-Medium benchmark). We generate these wall-clock estimates using a uniform setup using a single H100 GPU with 16 CPU cores of AMD EPYC 7742 processor. The results, summarized in the table below, confirm that multi-task BRC offers significant gains in efficiency even at reduced model scales. As discussed in Sections 4 and 5, these wall-clock improvements stem from improved sample efficiency paired with performing less gradient steps over the course of the training and are proportional to the number of tasks trained at once.
>
> |                                           | BRO (4M) | BRC (4M) | BRC (16M) | BRC (64M) | BRC (256M) |
> |-------------------------------------------|----------|----------|-----------|-----------|------------|
> | Final performance                         | 0.57     | 0.65     | 0.69      | 0.83      | 0.95       |
> | Minutes to perform 1M steps in all tasks  | 1013     | 242      | 256       | 336       | 658        |
> | Minutes to reach final performance of BRO | 1013     | 157      | 138       | 154       | 263        |
> | Wall-clock improvement                    | 1        | 6.4      | 7.3       | 6.6       | 3.9        |
>
> > The data transfer method sound more like JSRL...
>
> We thank the reviewer for the observation. We agree that our “data transfer” setup shares similarities with JSRL and WSRL as well. We have updated the relevant section to reflect this similarity. The key distinction between these approaches and "data transfer" is that in our approach, the agent is initialized with offline data from different tasks (e.g. solving humanoid-run using data from humanoid-stand), whereas in JSRL/WSRL/RLPD basic configuration, the offline data comes from the same target task. This cross-task transfer framing differentiates our setup from prior methods. **We’ve added this discussion to the revised manuscript**. If helpful for clarity, we’re also open to renaming “data transfer” in the final version to better align with existing terminology.
>
> > It seems like parameter scaling and task (data) scaling are not complementary...
>
> We appreciate the reviewer’s observation. While the narrowing gap between 3-task and 6-task performance in Figure 10 (right) is interesting, we do not believe it necessarily indicates that parameter scaling and data/task scaling are not complementary. One plausible interpretation is that, with enough training, the performance gap between generalist and specialist models on a given task naturally decreases. This phenomenon is also observed in LLMs: a generalist model pretrained on many tasks often performs well with minimal fine-tuning, but a smaller specialist model can eventually match or surpass its performance on a single task given sufficient data and training [1]. We also note that Figure 10 reflects only final performance at 1M steps, and does not capture differences in sample efficiency earlier in training (where we also see improvements). Finally, regarding results with the 9-task model: unfortunately, we cannot evaluate it on the same validation tasks used for the 1, 3 and 6 task settings, since the 9 task set includes those validation tasks and would introduce data leakage.
>
> [1] Hoffmann, Jordan, et al. "Training compute-optimal large language models." NeurIPS 2022
>
> > How would embedding-tuning transfer perform compared to other transfer methods?
>
> We thank the reviewer for this question. The “embedding tuning transfer” experiments were designed to explore the potential for sample-efficient adaptation and meta learning, rather than raw performance. Unlike “data transfer” and “model transfer” which involve full actor-critic training and require significant data (>500k transitions in our experiments), “embedding tuning transfer” freezes the actor-critic weights and only adjusts the task embedding vector using a very small amount of data (~300 transitions per task). Given this constraint, we expect embedding tuning to underperform in final performance compared to the more resource-intensive transfer methods. However, its key advantage lies in efficiency and rapid adaptation. We believe its potential would become more apparent in larger-scale settings, where rich task embeddings can be learned and reused across many tasks. We view this as a promising direction for few-shot RL, and plan to explore it further in future work.
>
> We thank the reviewer again for their helpful questions and suggestions. We hope that our rebuttal and implemented changes address all issues raised. If so, we kindly ask the reviewer to consider adjusting their score. We are also happy to further clarify or address new questions.

---

> > ### Comment · Reviewer_9y5s · 2025-08-02
> >
> > Thank you for the detailed response, I think my questions and concerns have been clarified. I will maintain my score.

---

> ### Author Response · Authors · 2025-08-02
> **Thank you for the rebuttal**
>
> Thank you for helping us to improve the paper. Please do not hesitate to post any other questions if they appear.

---

### Official Review · Reviewer_wYfY · 2025-07-01

**Clarity:** 2
**Significance:** 2
**Originality:** 2
**Rating:** 4
**Confidence:** 3

**Summary:**

The paper introduces BRC, a recipe that lets online value-based RL scale to billion-parameter models and hundreds of tasks without offline data or expert distillation. BRC combines three design choices: (1) a scaled, regularised residual critic (BroNet) that remains stable at large widths, (2) replacing MSE TD loss with cross-entropy via categorical Q-learning to equalise gradients across tasks with disparate reward scales, and (3) a single network conditioned on learnable task embeddings rather than separate heads. Multi-task BRC achieves comparable performance to task-specific specialists while performing up to 40 times fewer gradient steps, and the resulting networks transfer to unseen tasks with markedly better sample efficiency.

**Questions:**

1. Can BRC handle the observation noise, latency, and dynamics shift typical of physical robots?

2. Why does replacing the MSE TD loss with cross-entropy categorical Q-learning reduce gradient conflict across tasks?

3. How do BRC's gains scale when the critic is restricted to parameters less than 10M or when only one or two GPUs are available?

**Ethical Concerns:**

["NO or VERY MINOR ethics concerns only"]

**Limitations:**

Yes

**Quality:**

3

**Strengths And Weaknesses:**

***Strengths***:

1. BRC is evaluated on 283 tasks across five benchmarks. Robust ablations and Shapley analyses pinpoint the scaled BroNet critic as the main source of gains, lending credibility to the claim that careful architectural and loss design, rather than task-specific tuning, drives the performance.

2. The paper combines (i) a scaled, regularised residual Q-network (BroNet), (ii) cross-entropy TD loss via categorical Q-learning to equalise gradients across tasks, and (iii) learnable task embeddings in place of multiple heads. This triad is argued to act synergistically—removing any component degrades results.

***Weakness***:

1. Training billion-parameter critics on 283 tasks almost certainly requires very large GPU budgets, but the paper gives no wall-clock times, FLOPs, or carbon-footprint estimates. Without such data, reviewers cannot judge whether the reported efficiency (fewer gradient steps) translates into practical savings in energy or hardware.

2. It remains unclear whether resource-constrained settings (e.g., on-device robotics) reap any benefit, or whether BRC simply overpowers interference by brute force. Additional results at small/medium scales or a theoretical account of why cross-entropy stabilises TD learning would strengthen the contribution.

3. All benchmarks are simulators with well-shaped dynamics; no physical robot or real-data test is attempted. Moreover, although the paper briefly contrasts online vs. offline variants, deeper analysis of offline RL is missing.

---

> ### Author Rebuttal · Authors · 2025-07-31
>
> We thank the reviewer for helping us to improve the quality and impact of our work. We are happy that the reviewer found our experiments to be robust. To address reviewer’s concerns and further strengthen the manuscript, we propose the following changes:
>
> 1. We include new results comparing the wall-clock of BRC using different model sizes and BRO. These results demonstrate that multi-task BRC significantly reduces training time while improving the final performance.
> 2. We add experiments evaluating BRC under noisy observations and stochastic dynamics. Both single and multi-task BRC outperform BRO, highlighting the applicability of our approach to noisy tasks.
> 3. We have made a number of improvements to the manuscript based on reviewer feedback. These are detailed in our point-by-point responses.
>
> Please find our responses below:
>
> > Multi-task BRC achieves comparable performance to task-specific specialists
>
> As shown in Figures 1, 2, 4 and 9, multi-task BRC consistently outperforms single-task specialists across a wide range of tasks and specialists. To the best of our knowledge, this is the first work to demonstrate that a multi-task value-based RL training can lead to substantial performance improvements over its single-task counterpart in an online setting.
>
> > Training billion-parameter critics...
>
> While we present results for a 1B-parameter model in Figure 12, all core comparisons against baselines were conducted using a 250M parameter model to ensure reproducibility and accessibility. This detail is discussed in Appendices D and F, and **we clarify it more prominently in Section 4**, with explicit references to the relevant appendices. We hope that this change will improve the clarity of our work.
>
> > The paper gives no wall-clock times, FLOPs, or carbon-footprint estimates...
>
> We thank the reviewer for this suggestion. To better illustrate the practical efficiency of multi-task learning, **We now include wall-clock measurements** comparing efficiency of BRO and multi-task BRC across four model sizes for HumanoidBench (HB-Medium benchmark). We generate these wall-clock estimates using an uniform setup using a single H100 GPU with 16 CPU cores of AMD EPYC 7742 processor. The results, summarized in the table below, confirm that multi-task BRC offers significant gains in both performance and efficiency, even at reduced model scales, reinforcing its scalability and real-world applicability. As discussed in Sections 4 and 5, the wall-clock improvements stem from improved sample efficiency paired with performing less gradient steps over the course of the training and are proportional to the number of tasks trained at once.
>
> |                                        |BRO-4M|BRC-4M|BRC-16M|BRC-64M|BRC-256M|
> |----------------------------------------|------|------|-------|-------|--------|
> | Final performance                      | 0.57 | 0.65 | 0.69  | 0.83  | 0.95   |
> | Minutes to perform 1M steps (all tasks)| 1013 | 242  | 256   | 336   | 658    |
> | Minutes to reach BRO final performance | 1013 | 157  | 138   | 154   | 263    |
> | Wall-clock improvement                 | 1    | 6.4  | 7.3   | 6.6   | 3.9    |
>
> As follows, multi-task BRC offers wall-clock improvements, even when compared to BRO which uses 8x smaller batch size and 64x smaller critic. We also note that the FLOPs per gradient and simulation step are identical between the single and multi-task BRC (250M parameters, 1024 batch size, and simulation of all tasks). Therefore, the “40x improvement in gradient steps” can also be interpreted as a “40x improvement in FLOPs”. Our result highlights that the data-compute trade-off observed in supervised learning [1] persists value-based RL where multi-task training uses more diverse data to reduce compute cost per performance unit, especially in large-scale settings.
>
> [1] Kaplan, Jared, et al. “Scaling Laws for Neural Language Models”. arXiv
>
> > It remains unclear whether resource-constrained settings reap any benefit...
>
> We appreciate the reviewer’s point. To address the concern about resource-constrained settings, we note that several of our experiments (Figures 4, 5, and 10) are conducted using small models (<20M parameters). In particular, as shown in Figure 5 (left), our approach outperforms single-task specialists even when the critic model has fewer than 5M parameters (using ~20MB of memory). In all experiments we keep the actor network fixed at fewer than 200k parameters, incurring negligible memory overhead. This demonstrates that the benefits of BRC are not solely due to large-scale compute and that our method offers gains even under modest resource budgets. Furthermore, given the small size of the actor network, inference-time costs remain equivalent to SAC or PPO, making BRC practical for deployment scenarios such as on-device robotics. To better communicate this result to the reader, **we add a discussion of limitations of large models** to the limitations section, as well as **expand the discussion of performance with smaller models** in Section 5. We also do not believe that BRC overcomes interference simply through “brute force.” As shown in Figure 6 and as noted by the reviewer, all components contribute to performance gains, with removal of each component resulting in a statistically significant performance drop.
>
> > Theoretical account of why CE stabilises TD learning would strengthen the contribution
>
> We thank the reviewer for this suggestion. To make this clearer for readers, **we expand our discussion of previous works discussing the importance of loss functions in TD learning** [2,3] in Section 3.
>
> [2] Wang, Kaiwen, et al. "The benefits of being distributional: Small-loss bounds for reinforcement learning." NeurIPS
>
> [3] Wang, Kaiwen, et al. "The central role of the loss function in reinforcement learning." Statistical Science
>
> > No physical robot, deeper analysis of offline RL...
>
> We thank the reviewer for raising this point. Our goal in this work is to investigate whether RL models can effectively learn from diverse multi-task data in the online learning setting. Given resource constraints, we focused on simulation-based benchmarks, which allow controlled and repeatable evaluation across a wide range of tasks. To this end, we evaluated BRC across over 200 simulated tasks. We believe this constitutes one of the broadest evaluations in value-based RL to date, and provides a strong foundation for future works. Regarding offline RL, we agree that a deeper analysis would strengthen the paper. While our offline experiments are smaller than online counterparts, they suggest that our design principles also transfer to offline learning. We are particularly excited about applying BRC to real-world robotic learning, where sample efficiency and model transfer (as shown in Figure 10) are especially valuable and we plan to explore this direction in follow-up work. To point the reader to the potential of such experiments, **we expand the discussion of future work in the conclusion section**.
>
> > Can BRC handle noise...
>
> We thank the reviewer for this question. To evaluate BRC’s robustness to real-world conditions such as observation noise and stochastic dynamics, **we added a new experiment** using two environment wrappers: observation wrapper with noise ($\sigma$ = 0.05) added to each observation, simulating noisy state estimation; and action wrapper with noise ($\sigma$ = 0.05) applied to actions before they are executed, simulating stochastic or imprecise actuation. We summarize the results in the table below. Across both settings, BRC maintains strong performance and consistently outperforms BRO, indicating its robustness to moderate noise. We are happy to rerun this experiment with noise level that the reviewer finds appropriate.
>
> |                    |BRC(multi)|BRC(single)|BRO(single)|
> |--------------------|----------|-----------|-----------|
> | No noise           | 680      | 574       | 487       |
> | Noisy observations | 638      | 586       | 494       |
> | Noisy dynamics     | 417      | 373       | 338       |
>
> > Why does CE reduce gradient conflict across tasks?
>
> We thank the reviewer for this question. While our results in Figure 5 (right) suggest that critic model size and task embeddings have a primary impact on reducing gradient conflicts, we also observe that replacing the MSE with a CE contributes to stabilizing training dynamics in multi-task settings. Although we do not claim that CE loss directly minimizes gradient conflict, we show in Figure 3 that the CE leads to more consistent gradient norms across tasks, stabilizing learning. Overall, while CE may not directly align gradients across tasks, it contributes to a more stable optimization landscape. We see this as an important area for future theoretical and empirical analysis, and we hope our results help motivate such work.
>
> > How do BRC gains scale when restricted to less than 10M or when only one or two GPUs are available?
>
> We evaluate BRC performance across several critic sizes (1M, 4M, and 16M) in Figure 4. These results show that BRC delivers substantial performance gains even in low-resource settings. For example, for 1M parameters, multi-task BRC outperforms single-task SAC (denoted as "vanilla") by over 3x, and single-task BRO (denoted as "BroNet (MSE)") by over 2x. Regarding GPUs, all experiments in our paper were run on a single A100/H100 GPU, and no experiments required multi-GPU parallelism. We have also developed an optimized implementation of BRC that supports training on consumer-grade GPUs, and we will release this code to ensure accessibility.
>
> We thank the reviewer again for their helpful suggestions. We hope that the rebuttal clarified all issues, and that the added results strengthen our claims on the efficiency and practicality of BRC. If so, we kindly ask the reviewer to consider changing their score accordingly. Please inform us if we can further clarify any issues.

---

> ### Author Response · Authors · 2025-08-05
> **We thank the reviewer for their help**
>
> Thank you again for helping us to improve our manuscript. Could the reviewer please confirm whether the rebuttal clarifies all issues?

---

### Official Review · Reviewer_yjFR · 2025-07-03

**Clarity:** 3
**Significance:** 3
**Originality:** 2
**Rating:** 4
**Confidence:** 4

**Summary:**

The paper presents the BRC method, which scales value-based reinforcement learning (RL) models to achieve multi-task learning and higher efficiency both in terms of sample-wise and computation-wise. The paper proposes to leverage pure RL path rather than imitation learning or distillation to acquire multi-task policies by 1. cross-entropy based value learning; 2. high-capacity value function; 3. task-embedding. The authors did comprehensive and concrete experiments to demonstrate the efficacy of the proposed method. In conclusion, BRC enables scalable, efficient multi-task RL with high-capacity models and improved transfer learning.

**Questions:**

1. How do you define gradient conflicts? Why is it still high even in the single-task learning setting, as demonstrated in Fig.5 right?

**Ethical Concerns:**

["NO or VERY MINOR ethics concerns only"]

**Final Justification:**

Authors' rebuttals mainly resolved my concerns. They didn't claim SoTA performance on Atari, although they were selective on baselines.  Overall, I think this paper has been rigorously and comprehensively evaluated on common benchmarks, where the strong performance proved its efficacy. They also make clarifications on the definitions of gradient conflicts. I would recommend this paper to be accepted.

**Limitations:**

Yes.

**Paper Formatting Concerns:**

The authors do not provide the units for the x-axis of the performance figures in the Appendix. Although these are generally accepted by those in the field, they would affect the readability of the article.

**Quality:**

3

**Strengths And Weaknesses:**

Strengths:
1. Scaling value network has been a popular topic that hopes to make RL agents generalizable and perform better in multi-task settings. This paper is the first one to make impressive improvements on empirical results by conducting evaluations on almost all mainstream benchmarks.
2. Categorial learning + return normalization + task embedding combination is reasonable for multi-task design, especially for diverse reward margins.

Weakness:
1. This paper didn't include model-based RL algorithms into base models, which is expected because SAC+BRC has achieved stronger performance than the most advanced model-based RL algorithms on DMC. However, the author didn't take this into consideration.
2. Authors are selective about Atari baselines. It seems they exclude some of the SoTA model-based RL baselines like EfficientZero, which could lead to overclaiming about the improvements on the Atari-100k benchmark.

---

> ### Author Rebuttal · Authors · 2025-07-30
>
> We thank the reviewer for helping us to improve the quality and impact of our work. We are also very glad that the reviewer acknowledges the importance of our empirical results. To address the reviewer’s concerns and improve the quality of our work, we have made a few changes to our manuscript:
>
> 1. We discuss our focus on the model-free setting in the limitations section, clarifying that the scope of this work does not extend to model-based approaches due to our goal of understanding how the proposed design choices impact scalability of model-free RL
>
> 2. We clarify that our SOTA results are in the continuous control setting, which is the primary focus of our work, and do not extend to discrete control. The discrete control (Atari) experiments are included as secondary validation, meant to demonstrate that the same design principles generalize to multi-task learning in discrete domains, even when using a simple, compute-efficient baseline.
>
> 3. We expand our discussion on gradient conflict computation, including how we define and measure gradient conflict. These clarifications appear in Section 3 and the Appendix.
>
> 4. We add results measuring wall-clock performance of our proposed approach as compared to BRO. These results are supposed to better contextualize the efficiency improvements stemming from multi-task learning.
>
> Please find our detailed rebuttal below:
>
> > Additional results
>
> We would like to point the reviewer’s attention to newly added evaluations. To better ground our considerations on compute efficiency of multi-task learning, we now include wall-clock measurements comparing wall-clock times for BRO and multi-task BRC across four model sizes for HumanoidBench (HB-Medium benchmark). We generate these wall-clock estimates using an uniform setup using a single H100 GPU with 16 CPU cores of AMD EPYC 7742 processor. The results, summarized in the table below, confirm that multi-task BRC offers significant gains in efficiency even at reduced model scales. As discussed in Sections 4 and 5, these wall-clock improvements stem from improved sample efficiency, as well as performing less gradient steps over the course of the training and are proportional to the number of tasks trained at once.
>
> |                                           | BRO (4M) | BRC (4M) | BRC (16M) | BRC (64M) | BRC (256M) |
> |-------------------------------------------|----------|----------|-----------|-----------|------------|
> | Final performance                         | 0.57     | 0.65     | 0.69      | 0.83      | 0.95       |
> | Minutes to perform 1M steps in all tasks  | 1013     | 242      | 256       | 336       | 658        |
> | Minutes to reach final performance of BRO | 1013     | 157      | 138       | 154       | 263        |
> | Wall-clock improvement                    | 1        | 6.4      | 7.3       | 6.6       | 3.9        |
>
> > This paper didn't include model-based RL algorithms into base models...
>
> Our vision is to enable RL algorithms to successfully train large models on diverse multi-task datasets. We agree that the natural next step would be to extend our method to other settings, such as model-based or on-policy RL. For this study, however, we had to balance the scope with our compute budget, and we decided to focus on model-free settings. Model-free RL is a well-established setup, with previous works showing that multi-task learning requires techniques like gradient projection or distillation of single-task policies. This raises the question whether online value-based RL can successfully scale to many tasks at all without using multi-task-specific techniques. Our work shows that a simple algorithm can be used for both single and multi-task learning, with significant gains when used in a multi-task context even when compared to model-based TD-MPC2. Furthermore, we note that a lot of RL applications rely on a streamlined process in which a model is learned completely separately from the policy, and the policy is learned via a general model-free algorithm [1,2]. As such, we believe that our results are a good starting point for future works investigating model-based extensions to our approach. **We expanded our limitation section** to underscore that our results are restricted to the model-free setup. Furthermore, if the reviewer believes that adding model-based results would improve the reader’s experience, **we are happy to add basic experiments to the camera-ready version**.
>
> [1] Kaufmann, Elia, et al. "Champion-level drone racing using deep reinforcement learning." Nature 2023
>
> [2] Ouyang, Long, et al. "Training language models to follow instructions with human feedback." NeurIPS 2022
>
> > Authors are selective about Atari baselines...
>
> We thank the reviewer for pointing this out. Our primary focus is on demonstrating that training with diverse, multi-task data combined with specific design choices (cross-entropy, scaled critic, and task embeddings) improves value-based RL performance, especially in continuous action problems, where we do claim state-of-the-art results. In contrast, our Atari100k (discrete action) experiments serve as a secondary validation, aimed at testing the generality of our findings rather than achieving SOTA. To keep these experiments compute-efficient, we use a simple baseline (Efficient DQN + DrQ) and focus on comparing its multi-task and single-task variants. We acknowledge that we do not include stronger model-based baselines such as EfficientZero, and we agree that this limits the scope of our discrete-action claims. To make our intentions clear for the reader, **we now clarify this in the paper** by adding the following to the results section: “The results on the Atari100k benchmark show that the proposed design choices allow the agents to benefit from multi-task learning. While in our experiments we use the simple DrQ as base algorithm, combining these insights with approaches like BBF or EfficientZero might be a promising avenue for further improvement of these state-of-the-art algorithms.” We also add a note in the limitations section acknowledging that our discrete-action results are not competitive with the strongest existing baselines and serve as a proof of concept.
>
> > How do you define gradient conflicts? Why is it still high even in the single-task...
>
> We define gradient conflicts as in [3, page X], where they are defined by the cosine similarity between gradients stemming from different samples. Specifically, a conflict occurs when the cosine similarity is negative, indicating opposing gradient directions. When calculating gradient conflicts in multi-task setup, we calculate conflicts between transitions sampled from different tasks, whereas, in the single-task setup, we just divide the batch into two parts and calculate conflicts between these two subbatches. **We now clarify this definition in the main text** to avoid ambiguity. Regarding why gradient conflicts still appear in the single-task setting (as shown in Fig. 5, right): this stems from the observation that even within a single MDP, diverse data can give rise to competing optimization signals. As discussed in Section 3, using task embeddings can unify a multi-task setup into a single-task formulation by treating the multiple tasks as variations within a single MDP. For example, a point-mass agent with different goal locations can be considered a single-task MDP with high intra-task diversity, or a multi-task MDP where goal position can be encoded into a "task embedding". This intra-task diversity can lead to gradient conflicts if samples (e.g., reaching widely separated goals) produce gradients that point in conflicting directions. Therefore, gradient conflicts are not exclusive to multi-task learning and they can arise in single-task settings whenever the task exhibits sufficient structural diversity. Figure 5 empirically supports this by showing that the single-task setting still exhibits non-zero levels of gradient conflicts.
>
> [3] Yu, Tianhe, et al. "Gradient surgery for multi-task learning." NeurIPS 2020
>
>
> We thank the reviewer for their detailed feedback. Please let us know if the above rebuttal clarifies our results, especially with respect to the Atari benchmark. We also would like to point the reviewer’s attention to the added results on multi-task training efficiency, highlighting the practical impact of our work. Please let us know if there are any remaining issues. If not, we kindly ask the reviewer to consider adjusting their score.

---

> > ### Comment · Reviewer_yjFR · 2025-08-04
> >
> > Thanks for authors' clarifications. I understand that migrating BRC to a model-based RL algorithm may require extensive efforts, which is unrealistic considering the rebuttal timeframe. The authors expressed their willingness to improve the paper during the camera-ready phase.
> >
> > The responses for Atari experiments are not supportive enough for authors' claims, but in general this is still a high-quality paper.
> >
> > I think authors' responses resolve my concern for the definitions of gradient conficts. I would appreciate it if authors can further explain why the gradient conflict rate is such high even in single-task training.
> >
> > I prefer to keep my scores now.

---

> ### Author Response · Authors · 2025-08-04
> **We kindly ask the reviewer for further suggestions**
>
> Thank you for your response. We are glad that you think that the paper is high-quality already.
>
> Re the Atari results - We already added changes that explicitly say Atari results focus on **comparing single and multi-task in a ceteris paribus setting and should not be interpreted as SOTA results**. As such, we do not intend to "overclaim" with respect to the Atari results. If the reviewer has further reservations about our Atari results, we are happy to adjust the text further or change the presentation of these results according to the reviewer's recommendation. Please let us know.
>
> To further increase the reviewers' confidence in our work, we list the original results presented in our manuscript:
>
> 1. For the first time, we present an online multi-task RL method that significantly outperforms single-task specialists
>
> 2. For the first time, we show that a value model can be successfully transferred between different tasks in an online setting, leading to performance improvement
>
> 3. We validate our method using a total of >200 tasks corresponding to one of the bigger task sets in modern RL literature
>
> Thanks again for your time and feedback!

---

### Note · Authors · 2025-08-12

Dear AC and Reviewers,

Thank you for the time and effort you took to review our paper and provide feedback - we believe that the paper has improved significantly as a result. We summarize the most important changes to our manuscript below:

1. Following recommendations of Reviewers wYfY and 9y5s, we add a comparison of wall-clock efficiency showing that our proposed BRC leads to not only improvements in performance, but is also significantly faster (3x-7x faster than BRO-fast, while reaching 2x final performance)
2. Following the suggestion of Reviewer yjFR, we clarify that we propose an SOTA approach for continuous-action environments, while our results on Atari (discrete action) focus on ceteris paribus comparison of single and multi-task learning. Furthermore, we added an Appendix section in which we discuss the approach for calculating gradient conflicts, as well as comment on the gradient conflicts in single-task RL
3. Following the recommendations of Reviewer wYfY, we add an experiment that shows BRC robustness to observation and dynamics noise.
4. Following the suggestion of Reviewer zxeb, we add a paragraph in which we detail our approach to conditioning the actor and critic network on the task embedding vectors.

These results, as well as other results declared in the rebuttal, will be included in the final version of the paper.

Thank you so much!

---

### Decision · Program_Chairs · 2025-09-17

**Decision:**

Accept (poster)

**Comment:**

The paper proposes BRC, a simple and scalable recipe for online multi-task value-based RL combining: (1) a scaled, regularized residual critic (BroNet-style), (2) cross-entropy TD via categorical Q-learning with return normalization, and (3) learnable task embeddings instead of separate heads. The authors conduct a broad empirical study over 283 tasks spanning DMC, MetaWorld, HumanoidBench, ShadowHand, and Atari, using unified hyperparameters. Results show that multi-task BRC surpasses strong single-task specialists in continuous control with substantially fewer gradient steps; rebuttal-added wall-clock measurements further support practical efficiency. The work also demonstrates transfer to unseen tasks, including efficient embedding-only adaptation.

Strengths and weaknesses (largely addressed in rebuttal):
- Reviewers praise the breadth and rigor of experimentation, the clarity of writing, and the first convincing demonstration within online, value-based RL that multi-task training can surpass single-task specialists at scale.
- Concerns raised include modest conceptual novelty (the components are individually known), resource demands (large batch sizes and multi-task replay buffers), limited Atari baselines (authors have clarified that Atari is a proof-of-concept rather than SOTA), absence of real-robot experiments, and only limited exploration of model-based and offline RL. The authors’ rebuttal addresses gradient-conflict definitions, clarifies the Atari scope, and adds wall-clock comparisons and noise/stochasticity robustness results.

Overall, this is a good empirical and integrative contribution that meaningfully advances multi-task, value-based RL, showing that a simple, scalable recipe can outperform specialists and transfer effectively. Despite integrative novelty, the breadth, rigor, and practical impact justify acceptance. All four reviews are positive, and the rebuttal addressed key concerns. Please make sure the camera-ready incorporates reviewers' suggestions including scope and claims, compute and resources, return normalization and categorical support, and baselines and statistics.